# Mechanotransduction is required for establishing and maintaining mature inner hair cells and regulating efferent innervation

Laura F. Corns[1], Stuart L. Johnson[1], Terri Roberts[2], Kishani M. Ranatunga[2], Aenea Hendry[1], Federico Ceriani [1], Saaid Safieddine[3], Karen P. Steel[4], Andy Forge [5], Christine Petit[3,6], David N. Furness[7], Corné J. Kros[2] & Walter Marcotti [1]

In the adult auditory organ, mechanoelectrical transducer (MET) channels are essential for transducing acoustic stimuli into electrical signals. In the absence of incoming sound, a fraction of the MET channels on top of the sensory hair cells are open, resulting in a sustained depolarizing current. By genetically manipulating the in vivo expression of molecular components of the MET apparatus, we show that during pre-hearing stages the MET current is essential for establishing the electrophysiological properties of mature inner hair cells (IHCs). If the MET current is abolished in adult IHCs, they revert into cells showing electrical and morphological features characteristic of pre-hearing IHCs, including the re-establishment of cholinergic efferent innervation. The MET current is thus critical for the maintenance of the functional properties of adult IHCs, implying a degree of plasticity in the mature auditory system in response to the absence of normal transduction of acoustic signals.

[1] Department of Biomedical Science, University of Sheffield, Sheffield S10 2TN, UK. [2] School of Life Sciences, University of Sussex, Falmer, Brighton BN1 9QG, UK. [3] Unité de Génétique et Physiologie de l'Audition, Institut Pasteur, 75015 Paris, France. [4] Wolfson Centre for Age-Related Diseases, King's College London, London SE1 1UL, UK. [5] UCL Ear Institute, University College London, London SE1 1UL, UK. [6] Collège de France, 75005 Paris, France. [7] School of Life Sciences, Keele University, Keele ST5 5BG, UK. Correspondence and requests for materials should be addressed to W.M. (email: w.marcotti@sheffield.ac.uk)

The sense of hearing relies on sound evoking the displacement of stereociliary bundles projecting from the upper surface of hair cells. Hair bundle displacement causes the opening of mechanoelectrical transducer (MET) channels located at the tips of the shorter and middle rows of stereocilia, and the transduction of acoustic information into a receptor potential inside the hair cells. Stereocilia are interconnected by several extracellular linkages[1], including tip links that are directly involved in gating the MET channels. While tip links are formed by cadherin 23 (CDH23) and protocadherin 15 (PCDH15)[2], several other proteins are involved in tensioning and anchoring these links[3]. In the absence of sound, a fraction of the MET channels are normally open, allowing a standing depolarizing current into the hair cells that is mainly carried by $K^+$ and $Ca^{2+}$. Calcium entry through the MET channels causes adaptation and reduces their open probability[4]. However, the endolymph that surrounds the hair bundle has a low $Ca^{2+}$ concentration in both pre-hearing (second post-natal week: ~ 300 µM)[5] and adult (~ 20–40 µM)[6] cochleae, resulting in a larger in vivo resting open probability of the MET channel. The driving force for the MET current is provided by an electrical gradient between the endocochlear potential (pre-hearing: ~ + 20 mV; adult: ~ + 90 mV)[7] and the hair-cell resting potential (~ −60 mV)[8]. Although the resting MET current contributes to setting the membrane potential of pre-hearing and adult inner hair cells (IHCs)[5,8], its functional implication for IHC development is currently unknown.

Before the onset of hearing, IHCs fire spontaneous $Ca^{2+}$ action potentials (APs)[9,10], which play a crucial role in promoting neuronal survival[11] and the refinement of the immature auditory system[12,13]. Functionally mature IHCs respond to sound stimulation with graded receptor potentials. This change in the receptor potential requires a completely new complement of basolateral ion channels that appear just before the onset of hearing[9,14]. We hypothesize that, in the immature cochlea, the functional MET channels together with the maturing endocochlear potential[7] and endolymphatic $Ca^{2+}$ concentration[5], trigger the developmental switch from spiking IHCs to high-frequency signal transducers.

Using transgenic mice to manipulate the expression of molecular components of the MET apparatus in vivo, we demonstrate that the MET current is required for their developmental switch from immature to functional sensory receptors. Moreover, we found that the resting MET current in adult IHCs is required for maintaining their mature biophysical identity. The loss of the MET current in adult IHCs causes them to revert to an immature, pre-hearing phenotype and they re-acquire efferent innervation.

## Results

**MET channel failure prevents the maturation of IHCs.** Protocadherin 15 (PCDH15) is localized to the lower part of the tip link where it contacts the top of the shorter rows of stereocilia. Mutations in *Pcdh15* lead to hair-bundle disruption and loss of the MET current from early postnatal days[15]. Therefore, we investigated whether the absence of the MET current in PCDH15 deficient mice affected hair cell maturation. IHCs acquire most of their mature biophysical and morphological characteristics around the onset of hearing (~ P12). This includes the expression of a rapidly activating large conductance $Ca^{2+}$-activated $K^+$ current ($I_{K,f}$)[9] and a $K^+$ current ($I_{K,n}$)[14] carried by KCNQ4 channels. We found that $I_{K,f}$ was present in adult control IHCs (*Pcdh15*$^{+/av3J}$: Fig. 1a, 3.8 ± 0.5 nA at −25 mV, $n = 10$, mean ± SEM; see Methods for details) but absent in *Pcdh15*$^{av3J/av3J}$ mutant mice (Fig. 1b). The absence of $I_{K,f}$ in *Pcdh15*$^{av3J/av3J}$ unmasked the inward $Ca^{2+}$ current (−252 ± 23 pA, $n = 12$). The size of $I_{K,n}$ was significantly reduced in *Pcdh15*$^{av3J/av3J}$ IHCs (−68 ± 11 pA at −124 mV, $n = 8$, $P < 0.0001$, t-test; see

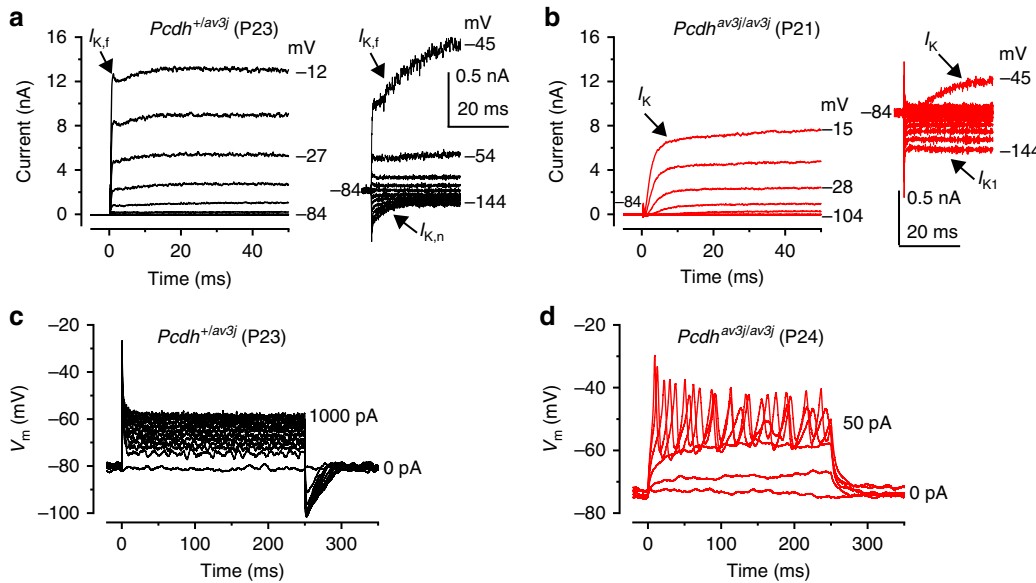

**Fig. 1** Current and voltage responses from IHCs of *Pcdh15* mutant mice. **a**, **b** Potassium currents recorded from IHCs of control *Pcdh15*$^{+/av3J}$ (**a**) and littermate mutant *Pcdh15*$^{av3J/av3J}$ mice (**b**) using 10 mV depolarizing voltage steps from −84 mV to the various test potentials shown by some of the traces. The adult-type currents ($I_{K,f}$ and $I_{K,n}$) were only present in IHCs from control *Pcdh15*$^{+/av3J}$ mice (**a**). IHCs from *Pcdh15*$^{av3J/av3J}$ mice retained the currents characteristic of immature cells (**b**, $I_K$ and $I_{K1}$). The absence of the rapidly activating $I_{K,f}$ in *Pcdh15*$^{av3J/av3J}$ IHCs is also evident when comparing the activation time course of the total outward currents on an expanded time scale (see insets). **a**, **b** are representative recordings from 10 (3 mice) and 12 (3 mice) IHCs, respectively. **c**, **d** Voltage responses elicited by applying depolarizing current injections to control, *Pcdh15*$^{+/av3J}$ **c** and mutant, *Pcdh15*$^{av3J/av3J}$ **d** IHCs from their resting membrane potentials. Depolarizing current injections caused slow APs in the *Pcdh15*$^{avj3/av3J}$ mutant IHC. **c**, **d** are representative recordings from 8 (3 mice) and 15 (3 mice) IHCs, respectively. Recordings were performed near body temperature

Methods for details) compared to that of $Pcdh15^{+/av3J}$ control cells ($-216 \pm 17$ pA, $n = 8$). $Pcdh15^{av3J/av3J}$ IHCs retained an immature phenotype by expressing the inward ($I_{K1}$) and slowly-activating outward ($I_K$) K$^+$ currents[14,16]. The size of $I_K$, calculated as the difference between the steady-state and early ($I_{K,f}$) current was $2.4 \pm 0.5$ nA ($Pcdh15^{+/av3J}$) and $1.6 \pm 0.2$ nA ($Pcdh15^{av3J/av3J}$) ($P = 0.159$, $t$-test). The physiological consequence of the immature current profile in adult $Pcdh15^{av3J/av3J}$ IHCs was that, instead of generating rapid and graded voltage responses characteristic of adult IHCs (Fig. 1c), they retained the ability to fire action potentials (APs) (Fig. 1d), a characteristic of pre-hearing IHCs[14]. A similar failure in the acquisition of the mature current profile has also been reported in IHCs from mice lacking other stereociliary bundle proteins, including the transmembrane channel-like protein 1 (TMC1) (Deafness mouse)[17], which is necessary for mechanoelectrical transduction[18], and the unconventional Myo6 (Snell's waltzer mouse)[19], which is required for the development of the MET apparatus[20]. The above findings indicate the existence of a link between a functional MET apparatus and IHC maturation. Mutations in or absence of stereociliary proteins early in development normally causes severe morphological defects of the stereociliary bundle (e.g. $Myo6$;[21] $Myo7a$;[22] harmonin[23]). We used targeted knockouts obtained by crossing floxed mice ($Ush1c^{fl/fl}$ and $Myo7a^{fl/fl}$) with $Myo15$-cre$^{+/-}$ mice[24]. Because the temporal expression of the $cre$ driven by the hair cell specific $Myo15$ promoter occurs postnatally ($\sim$P4 in the cochlear apical coil)[24], the stereociliary bundle develops normally, allowing us to study the functional implications of the missing protein.

**The MET current drives IHC maturation.** Harmonin is a scaffolding protein that contributes to the attachment of the upper end of the tip link to the actin filament of the stereocilia[25]. In $Ush1c$ (harmonin) mutant mice the MET channel lacks the resting current and by P7 the MET current is almost completely absent[23]. Although the harmonin-a isoform has been implicated in Ca$^{2+}$ channel regulation, this role is restricted to post-hearing IHCs[26]. Therefore, we recorded MET currents from $Ush1c^{fl/fl}$ and $Ush1c^{fl/fl}Myo15$-cre$^{+/-}$ immature IHCs (Fig. 2) by displacing the hair bundles[4]. At P7 the size of the maximum MET current was similar between the two genotypes ($Ush1c^{fl/fl}$: $-1117 \pm 83$ pA, $n = 4$; $Ush1c^{fl/fl}Myo15$-cre$^{+/-}$: $-1043 \pm 311$ pA, $n = 2$, measured at $-121$ mV, $P = 0.763$, $t$-test, Fig. 2a, b). The absence of a clear phenotype at the end of the first postnatal week agrees with previous observations using the $Myo15$-cre mouse to knockdown another stereociliary protein (sans)[24]. However, at P8 (Fig. 2b) the size of the MET current was already significantly smaller in $Ush1c^{fl/fl}Myo15$-cre$^{+/-}$ ($-828 \pm 77$ pA, $n = 3$, $P < 0.04$, $t$-test) compared to control cells ($Ush1c^{fl/fl}$: $-1079 \pm 61$ pA, $n = 6$), and it deteriorated rapidly thereafter such that by P10 it was almost absent in $Ush1c^{fl/fl}Myo15$-cre$^{+/-}$ IHCs (Fig. 2c, d; $Ush1c^{fl/fl}$ $-676 \pm 41$ pA, $n = 9$; $Ush1c^{fl/fl}Myo15$-cre$^{+/-}$ $-73 \pm 30$ pA, $n = 17$, P10-P12, $P < 0.0001$, $t$-test). The fraction of the MET channel open at rest in P10-P12 IHCs was negligible in $Ush1c^{fl/fl}Myo15$-cre$^{+/-}$ mice (at $-121$ mV: $0.003 \pm 0.001$; at $+99$ mV: $0.010 \pm 0.007$, $n = 17$) and significantly reduced compared to that of control cells ($Ush1c^{fl/fl}$ at $-121$ mV: $0.036 \pm 0.008$, at $+99$ mV: $0.120 \pm 0.034$, $n = 6$, $P < 0.0001$, $t$-test). Despite the abnormal MET current, the hair bundles of $Ush1c^{fl/fl}Myo15$-cre$^{+/-}$ IHCs acquired the characteristic staircase structure and appeared qualitatively similar to littermate controls at P15 (Fig. 2e). Using TEM, we quantified the height and width of the tallest (s1) and second tallest (s2) stereocilia in P25 IHCs. Although the height and width of s1 were not significantly different between the two genotypes ($P < 0.7068$ and $P < 0.3788$, $t$-test, respectively, Fig. 2f),

the former showed greater variability in $Ush1c^{fl/fl}Myo15$-cre$^{+/-}$ IHCs. The height and width of s2 for $Ush1c^{fl/fl}Myo15$-cre$^{+/-}$ IHCs ($1.42 \pm 0.14$ $\mu$m and $0.35 \pm 0.07$ $\mu$m, respectively, mean $\pm$ SD, $n = 7$ IHCs) were significantly reduced compared to control cells ($1.78 \pm 0.14$ $\mu$m, $P < 0.0005$; $0.45 \pm 0.03$ $\mu$m, $P < 0.0066$, $t$-test, $n = 7$ IHCs, Fig. 2f). The reduced s2 compared to s1 in $Ush1c^{fl/fl}Myo15$-cre$^{+/-}$ IHCs, meant that the s1/s2 ratio was increased ($3.16 \pm 0.57$) compared to control cells ($2.37 \pm 0.12$, $P < 0.0036$, $t$-test). Finally, s2 in IHCs from $Ush1c^{fl/fl}Myo15$-cre$^{+/-}$ mice tended to show rounded tips rather than the characteristic prolate-shaped stereocilia tips of controls (Fig. 2f), which strongly suggests that the tip links are absent. Although some prolate-shaped stereocilia tips were present is s2 at P15 in $Ush1c^{fl/fl}Myo15$-cre$^{+/-}$ mice, they were less evident than those in controls (Supplementary Fig. 1), indicating that the loss of the MET current in the absence of $Ush1c$ could be due to the progressive loss of tip links.

A determining factor for the presence of spontaneous AP activity in pre-hearing IHCs (second post-natal week) is the resting membrane potential ($V_m$), which has to be close to the activation threshold of the Ca$^{2+}$ current. In IHCs $V_m$ is determined by the balance between an inward, depolarizing resting MET current and outward $I_K$[5]. Therefore, we investigated whether the substantially reduced resting MET current in immature IHCs from $Ush1c^{fl/fl}Myo15$-cre$^{+/-}$ mice affected IHC physiology. Since the fraction of MET channels open at rest is determined by Ca$^{2+}$-dependent adaptation[4,27], the $V_m$ of IHCs in vitro strongly depends upon the extracellular Ca$^{2+}$ concentration. Using current clamp, we measured the depolarizing current required to elicit APs in IHCs at 35–37 °C using 1.3 mM Ca$^{2+}$, which is the Ca$^{2+}$ concentration present in the perilymphatic solution bathing the IHC basolateral membrane. Starting from the resting $V_m$ ($Ush1c^{fl/fl}$: $-72.2 \pm 2$ mV; $Ush1c^{fl/fl}Myo15$-cre$^{+/-}$: $-74.9 \pm 2$ mV, $P = 0.257$, $t$-test), a similar amount of current was required to trigger APs in both control $Ush1c^{fl/fl}$ ($75 \pm 7$ pA, $n = 10$, P9-P10) and $Ush1c^{fl/fl}Myo15$-cre$^{+/-}$ ($72 \pm 9$ pA, $n = 13$, P9-P11, $P = 0.770$, $t$-test) mice (Fig. 2g). We then measured the resting MET current near the $V_m$ of IHCs in 1.3 mM Ca$^{2+}$ and at room temperature, and found that it was much smaller (Fig. 2h) than that required to elicit APs (Fig. 2g), even when correcting for body temperature (Fig. 2h). However, the degree of adaptation of the MET channel in vivo should be much reduced as the estimated endolymphatic Ca$^{2+}$ concentration during the second postnatal week is 0.3 mM[5]. The application of 0.3 mM Ca$^{2+}$ increased the resting MET current in control $Ush1c^{fl/fl}$ IHCs (Fig. 2i, j) to a value that would depolarize them to their AP firing threshold (Fig. 2g: also see ref. [5]), and even more so when values were adjusted for the temperature difference (Fig. 2i). This would also be the case if we were to subtract the resting MET current measured in 1.3 mM Ca$^{2+}$ (Fig. 2h), which is likely to contribute to the resting $V_m$ and thus to the current required for eliciting repetitive firing (Fig. 2g). The reduced resting MET current in $Ush1c^{fl/fl}Myo15$-cre$^{+/-}$ IHCs bathed in 0.3 mM Ca$^{2+}$(Fig. 2i) would not be sufficient to depolarize them to AP threshold. IHCs however would still be able to fire APs in response to ATP-induced Ca$^{2+}$ waves from surrounding non-sensory cells[10,28].

We then investigated whether the absence of harmonin from pre-hearing $Ush1c^{fl/fl}Myo15$-cre$^{+/-}$ IHCs would also affect their basolateral membrane properties. At P7, P9, and P10, the latter being a time when the MET current in mutant IHCs is almost absent (Fig. 2c, d), the biophysical properties of IHCs, including the size of K$^+$ currents, resting membrane potential and cell membrane capacitance, were not significantly different between $Ush1c^{fl/fl}$ and $Ush1c^{fl/fl}Myo15$-cre$^{+/-}$ mice (Supplementary Table 1, Supplementary Fig. 2). The above data show that the loss of

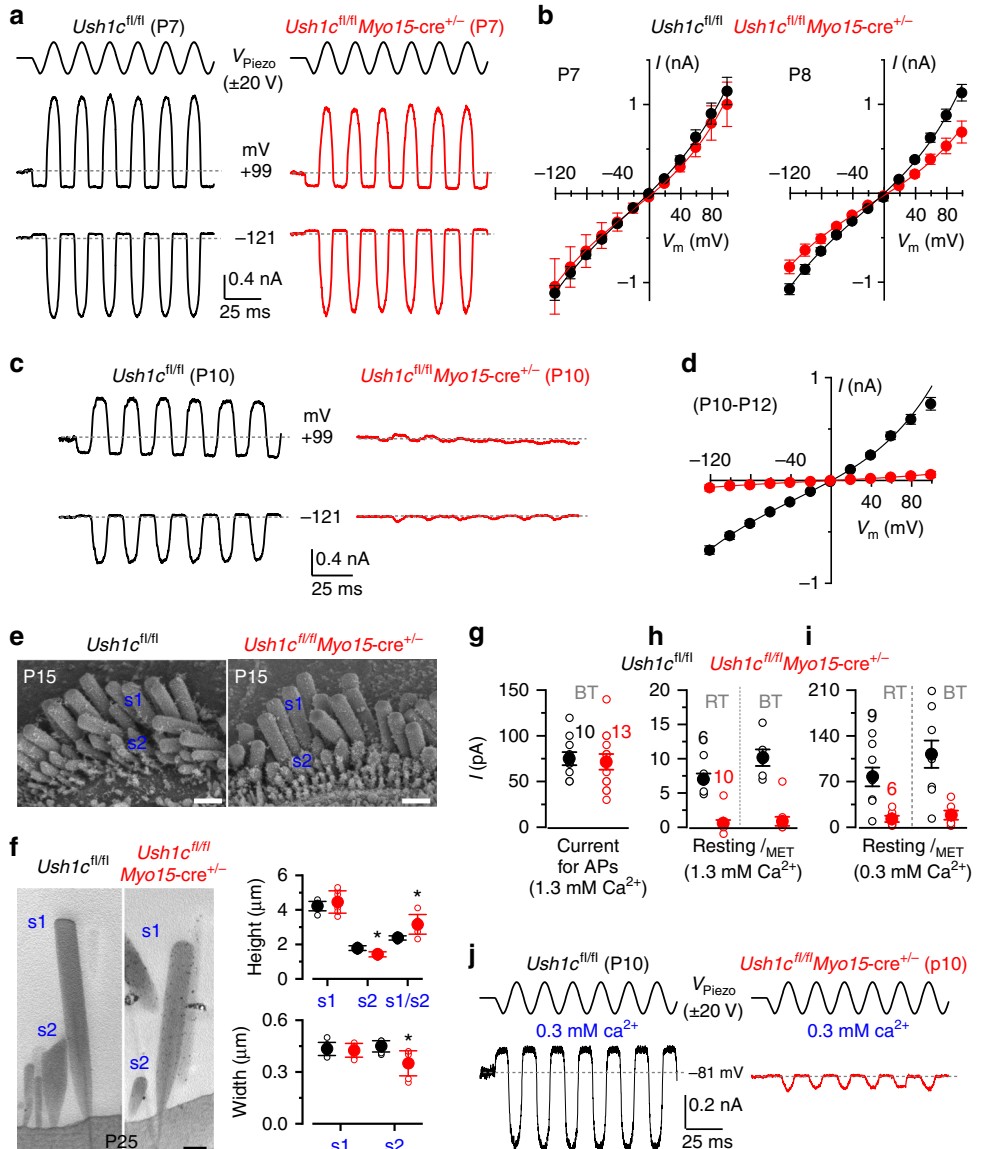

**Fig. 2** Disappearance of the MET current in IHCs of *Ush1c*fl/fl conditional KO mice. **a** MET currents recorded in P7 IHCs from control *Ush1c*fl/fl and knockout *Ush1c*fl/fl*Myo15*-cre+/− mice by displacing the hair bundles while applying 20 mV voltage steps between −121 mV to +99 mV. For clarity, only the traces at −121 mV and +99 mV are shown (dashed lines: holding current; $V_{Piezo}$: driver voltage to the fluid jet). **b** Maximum MET current-voltage curves obtained from P7 (left) and P8 (right) *Ush1c*fl/fl and *Ush1c*fl/fl*Myo15*-cre+/− IHCs (3 and 6 mice, respectively). **c** MET currents in P10 IHCs from both genotypes. Protocol as in **a**. **d** Maximum MET currents obtained from 9 *Ush1c*fl/fl and 17 *Ush1c*fl/fl*Myo15*-cre+/− (P10-P12) IHCs (4 and 8 mice, respectively). **e** SEM images of P15 IHC bundle from both genotypes. Scale bars, 1 μm. **f** TEM images (left) of P25 IHC stereocilia from both genotypes. Scale bars, 400 nm. Right panels show the height and width of the tallest (s1) and second tallest (s2) stereocilia in P25 IHCs from *Ush1c*fl/fl and *Ush1c*fl/fl*Myo15*-cre+/− mice. **g** Depolarizing current used to elicit repetitive APs in *Ush1c*fl/fl (P9-P10, 8 mice) and *Ush1c*fl/fl*Myo15*-cre+/− (P9-P11, 7 mice) IHCs. Experimental conditions: body temperature (BT: 34–37 °C); 1.3 mM extracellular $Ca^{2+}$. **h** Size of the resting MET current ($I_{MET}$) recorded from *Ush1c*fl/fl IHCs (P9-P11, 2 mice) and *Ush1c*fl/fl*Myo15*-cre+/− (P10-P11, 5 mice) IHCs. Room temperature (RT: 22–24 °C, left bars); 1.3 mM extracellular $Ca^{2+}$. $I_{MET}$ was interpolated from its current-voltage relation near the $V_m$ of IHCs (−73 mV). Right bars show the RT values corrected for BT using a $Q_{10}$ of 1.45 (ref. [8]). **i** Size of the resting $I_{MET}$ at −73 mV from *Ush1c*fl/fl (P10, 3 mice) and *Ush1c*fl/fl*Myo15*-cre+/− (P10, 3 mice) IHCs using the endolymph-like $Ca^{2+}$ concentration (0.3 mM: see panel **j**). Left and right bars are as described in **h**. **j** $I_{MET}$ recorded at −81 mV at P10 from *Ush1c*fl/fl and *Ush1c*fl/fl*Myo15*-cre+/− IHCs in 0.3 mM $Ca^{2+}$. Protocol as in **a**. Number of IHCs in **g**–**i** is shown above the columns. Average data is shown as mean ± SEM in **b**, **d**, **g**, **h** or mean ± SD in **f**

harmonin leads to a gradual disappearance of the MET current without affecting normal development of the basolateral membrane properties of pre-hearing IHCs, including the $Ca^{2+}$ channels[26]. After the onset of hearing, while young adult (P14-P15) and adult (P29) IHCs from control *Ush1c*fl/fl mice express both $I_{K,f}$ and $I_{K,n}$ (Fig. 3a) and exhibit graded voltage responses (Fig. 3c), age-matched *Ush1c*fl/fl*Myo15*-cre+/− IHCs retained an immature current profile (Fig. 3b, Supplementary Table 2) and the ability to generate

APs (Fig. 3d). Auditory brainstem responses (ABRs) showed that *Ush1c*fl/fl*Myo15*-cre+/− mice had severe hearing impairment at P16 (Fig. 3e) and were profoundly deaf at P37-45 (Fig. 3f). Adult mice also exhibit the classical head bobbing and hyperactive circling behavior characteristic of *Ush1c* mutant mice[29]. The above results indicate that the acquisition of the mature basolateral membrane profile in IHCs requires a functional MET current during pre-hearing stages of development.

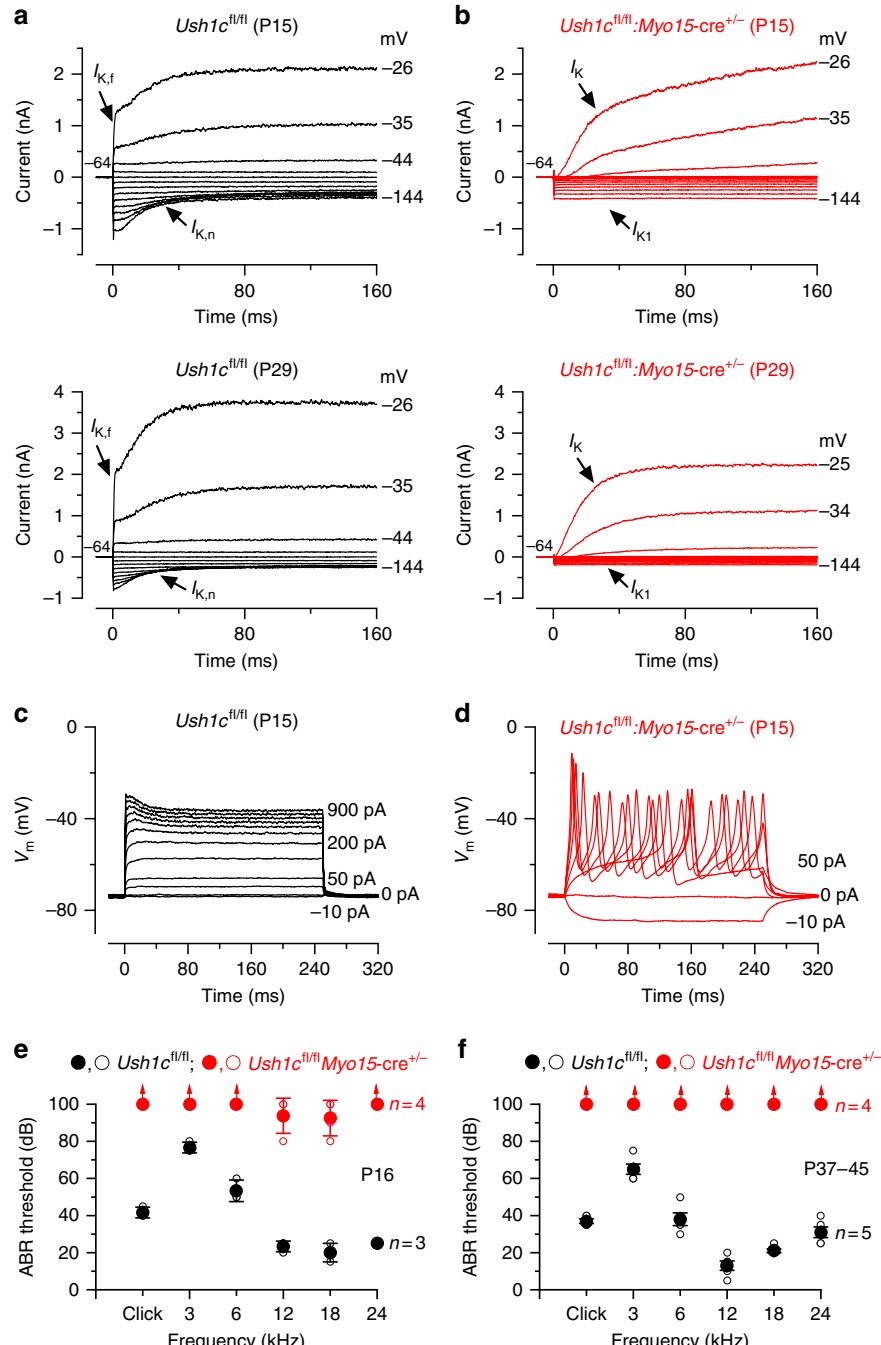

**Fig. 3** IHCs from $Ush1c^{fl/fl}Myo15$-cre$^{+/-}$ conditional knockout mice fail to mature. **a, b** Potassium currents recorded from P15 (upper panels) and P29 (lower panels) IHCs from control $Ush1c^{fl/fl}$ (**a**) and $Ush1c^{fl/fl}Myo15$-cre$^{+/-}$ (**b**) mice using 10 mV depolarizing and hyperpolarizing voltage steps from −64 mV. Note that the K$^+$ currents characteristic of adult IHCs ($I_{K,f}$ and $I_{K,n}$) were only present in IHCs from control $Ush1c^{fl/fl}$ mice (**a**). IHCs from $Ush1c^{fl/fl}Myo15$-cre$^{+/-}$ mice retained the K$^+$ currents characteristic of pre-hearing stages ($I_K$ and $I_{K1}$). **c, d** Voltage responses elicited by applying hyperpolarizing and depolarizing current injections from the respective membrane potentials in P15 IHCs of control $Ush1c^{fl/fl}$ (**c**) and $Ush1c^{fl/fl}Myo15$-cre$^{+/-}$ (**d**) mice. Note that APs are still present in post-hearing $Ush1c^{fl/fl}Myo15$-cre$^{+/-}$ IHCs. For more details see Supplementary Table 2. **e, f** Mean ABR thresholds ( ±SD ) for clicks and pure tone stimulation from 3 kHz to 24 kHz obtained from control $Ush1c^{fl/fl}$ and littermate $Ush1c^{fl/fl}Myo15$-cre$^{+/-}$ mice at P16 (**e**) and P37-45 (**f**). $Ush1c^{fl/fl}Myo15$-cre$^{+/-}$ mice are almost completely deaf already at P16. Open symbols represent data from single mice. Closed symbols represent averages (mean ± SEM)

**Mature IHCs regress without the MET current**. The unconventional myosin VIIa (Myo7a) is another hair cell protein that binds to harmonin[30]. In the absence of Myo7a no MET current is present at rest[31], stereociliary bundles gradually degenerate ($Myo7a^{sh6J/sh6J}$ mice)[31] and the K$^+$ current profile of adult IHCs retains pre-hearing characteristics (Supplementary Fig. 3). Different from pre-hearing $Ush1c^{fl/fl}Myo15$-cre$^{+/-}$ mice (Fig. 2),

the size (−648 ± 40 pA at −121 mV, $n = 6$, 3 mice, P9-P10) and resting open probability (0.026 ± 0.004 pA, $n = 6$) of the MET current in IHCs of pre-hearing $Myo7a^{fl/fl}Myo15$-cre$^{+/-}$ mice was similar to that measured in control $Myo7a^{fl/fl}$ IHCs (−725 ± 38 pA, $P = 0.225$, $t$-test; 0.038 ± 0.010 pA, $P = 0.2318$, $t$-test, $n = 4$, 3 mice, P10) (Fig. 4a, b). The normal MET current in pre-hearing $Myo7a^{fl/fl}Myo15$-cre$^{+/-}$ IHCs (Fig. 4a, b), but not in

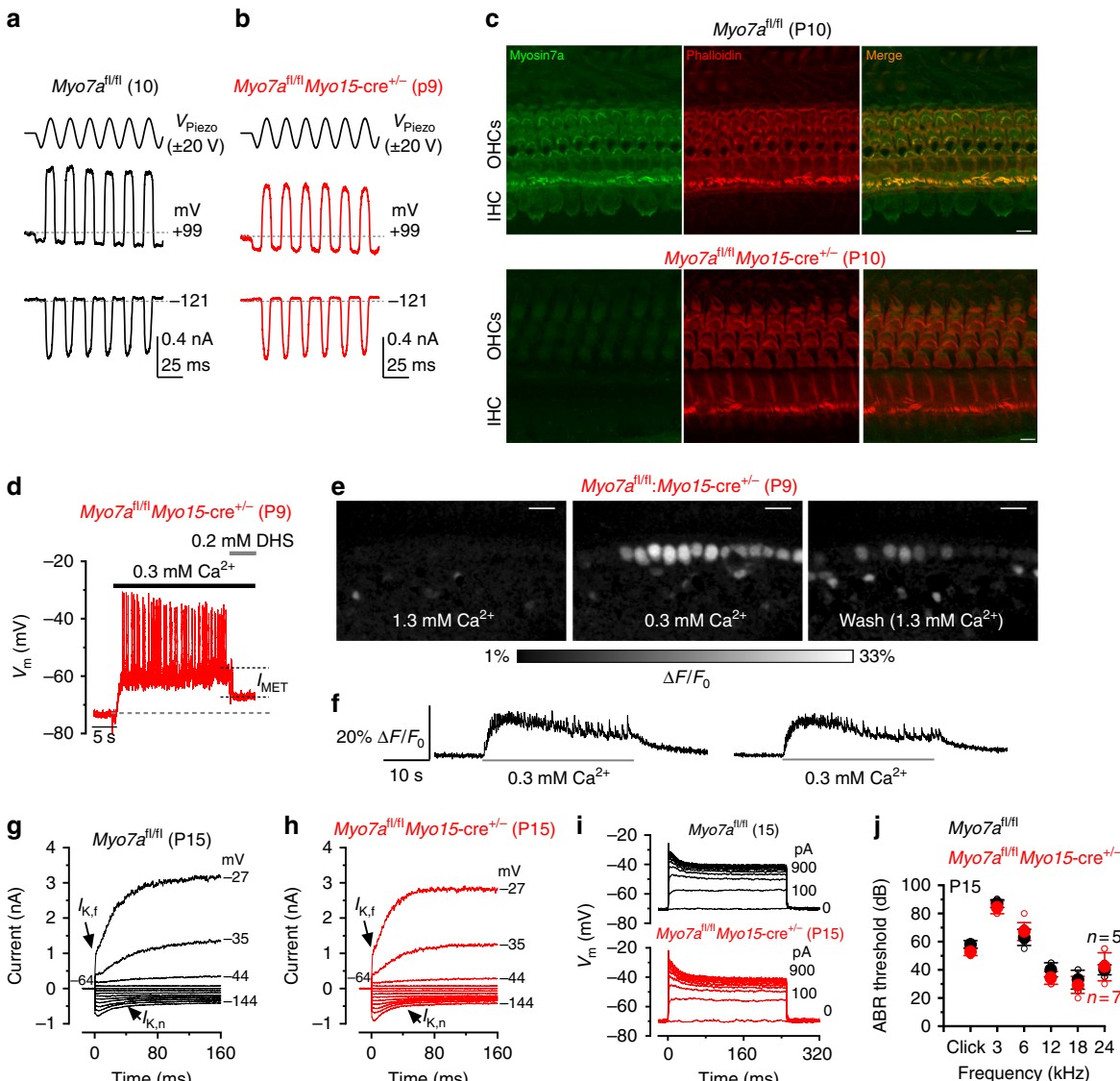

**Fig. 4** IHCs from $Myo7a^{fl/fl}Myo15$-cre$^{+/-}$ conditional knockout mice acquire their mature phenotype. **a**, **b** MET currents from IHCs of control $Myo7a^{fl/fl}$ (left, P10) and knockout $Myo7a^{fl/fl}Myo15$-cre$^{+/-}$ (right, P9) mice. The experimental protocol is as in Fig. 2. **c** Cochlear whole mount preparations from P10 $Myo7a^{fl/fl}$ (control, upper panel) and $Myo7a^{fl/fl}Myo15$-cre$^{+/-}$ (lower panel) mice, immunostained for Myo7a (green) and for F-actin with phalloidin (red). Myo7a is detected in both OHCs and IHCs in the $Myo7a^{fl/fl}$ mouse, but not in the $Myo7a^{fl/fl}Myo15$-cre$^{+/-}$ mouse (but see also Supplementary Fig. 4 for one example of normal hair cell Myo7a labeling in a P10 $Myo7a^{fl/fl}Myo15$-cre$^{+/-}$ mouse). Scale bars, 5 μm. **d** IHC voltage response recorded during the superfusion of 0.3 mM Ca$^{2+}$ alone and together with the MET channel blocker DHS (0.2 mM) in a $Myo7a^{fl/fl}Myo15$-cre$^{+/-}$ mouse. The MET depolarizing current substantially contributes to the IHC membrane potential. **e** Images showing relative fluo-4 fluorescence changes ($\Delta F/F_0$) before (left) during (middle) and after (right) application of an extracellular solution containing 0.3 mM Ca$^{2+}$. Images were obtained as maximal back projection (200 frames, 6.6 s). Scale bar, 15 μm. **f** Representative $\Delta F/F_0$ traces from two IHCs (from images in panel **e**) during bath application of 0.3 mM Ca$^{2+}$. Traces are computed as pixel averages of regions of interest centered on IHCs. Ca$^{2+}$ spikes are evident in IHCs; however, the relative long fluorescence decay time constant of the Ca$^{2+}$ signal, prevents the isolation of single APs during high-frequency bursting. **g**, **h** Current responses in post-hearing IHCs (P15) from control (**g**) and $Myo7a^{fl/fl}Myo15$-cre$^{+/-}$ (**h**) mice. Note that both $I_{K,f}$ and $I_{K,n}$ are present in IHCs from both genotypes. Voltage protocol as in Fig. 3a. **i** Voltage responses recorded from P15 IHCs of control $Myo7a^{fl/fl}$ (top panel) and $Myo7a^{fl/fl}Myo15$-cre$^{+/-}$ (lower panel) P15 IHCs (protocol as in Fig. 3c). Panel (**g**, **j**: $Myo7a^{fl/fl}$) and (**b**, **j**: $Myo7a^{fl/fl}Myo15$-cre$^{+/-}$) are representative recordings from 9 IHCs (4 mice) and 8 IHCs (4 mice), respectively. **j** Mean ABR thresholds (± SD) for clicks and frequency-specific pure tone stimulation from 3 Hz to 24 kHz obtained from control $Myo7a^{fl/fl}$ and littermate $Myo7a^{fl/fl}Myo15$-cre$^{+/-}$ mice at P15

$Ush1c^{fl/fl}Myo15$-cre$^{+/-}$ cells (Fig. 2), suggests a possible slower turnover of Myo7a, compared to that of harmonin. This is supported by the normal appearance of Myo7a immunostaining in hair cells from one out of four P10 mice tested (Fig. 4c; Supplementary Fig. 4). Alternatively, the late phenotype in $Myo7a^{fl/fl}Myo15$-cre$^{+/-}$ IHCs could reflect a distinct, and as yet undefined, role for Myo7a in mechanoelectrical transduction.

The aforementioned experiments (Fig. 2g–j) show that the resting MET current is required for the generation of spontaneous APs in IHCs during the last days preceding the onset of hearing. The normal MET current in IHCs of $Myo7a^{fl/fl}Myo15$-cre$^{+/-}$ mice meant that superfusion of the estimated endolymphatic Ca$^{2+}$ concentration (0.3 mM Ca$^{2+}$)[5], caused them to depolarize to near the expected in vivo resting $V_m$ and elicited repetitive APs

(Fig. 4d–f). The incomplete repolarization of IHC by the MET channel blocker dihydrostreptomycin (DHS) is due to the low $Ca^{2+}$ solution additionally affecting basolateral membrane SK2 channels through the small hole in the epithelium created to gain access to the cell with the patch pipette[5]. Indeed, two-photon confocal $Ca^{2+}$ imaging experiments, in which the sensory epithelium is kept intact, showed that 0.3 mM $Ca^{2+}$ caused IHCs to fire $Ca^{2+}$ spikes superimposed on a sustained increase in fluorescence (Fig. 4e, f), confirming that the MET current-induced IHC depolarization by endolymphatic $Ca^{2+}$ is sufficient to generate repetitive APs. The normal development of IHCs in $Myo7a^{fl/fl}Myo15$-cre$^{+/-}$ mice meant that post-hearing cells (P14-P15) acquired current and voltage responses similar to control $Myo7a^{fl/fl}$ cells (Fig. 4g–i; Supplementary Table 3), further supporting the hypothesis that a normal resting MET current, and the associated AP activity, is required for the final IHC functional differentiation. Moreover, at P15, $Myo7a^{fl/fl}Myo15$-cre$^{+/-}$ mice show ABRs comparable to those of control mice (click thresholds: $Myo7a^{fl/fl}$ 58 ± 3 dB, $n = 5$; $Myo7a^{fl/fl}Myo15$-cre$^{+/-}$ 53 ± 3 $n = 7$; Fig. 4j). ABR thresholds at all frequencies tested were also similar between the two genotypes (two-way ANOVA, $P = 0.174$).

Despite normal ABR thresholds in P15 $Myo7a^{fl/fl}$ $Myo15$-cre$^{+/-}$ mice, the MET current became progressively affected from P14 compared to control $Myo7a^{fl/fl}$ IHCs (Fig. 5a–c). The size of the MET current and the fractional resting open probability of the channel in control IHCs ($Myo7a^{fl/fl}$: $-974 ± 63$ pA, 0.108 ± 0.028, $n = 6$, respectively, measured at $-121$ mV, P15) were significantly larger than those from $Myo7a^{fl/fl}Myo15$-cre$^{+/-}$ cells at P14 ($-669 ± 96$ pA, $n = 4$, $P < 0.05$; 0.011 ± 0.014, $P < 0.05$) and P16 ($-377 ± 64$ pA, $n = 6$, $P < 0.001$; 0.031 ± 0.014, $P < 0.05$) (one-way ANOVA). The smaller MET current in $Myo7a^{fl/fl}$ $Myo15$-cre$^{+/-}$ IHCs after the onset of hearing occurred despite the normal appearance of the stereociliary bundles and the presence of tip links as well as prolate-shaped stereocilia tips (P30: Fig. 5d). Using TEM we found that the height and width of stereocilia in the tallest (s1) and second tallest (s2) rows in P37 IHCs were not significantly different between the two genotypes (Fig. 5e). The gradual loss of MET current was followed by a progressive disappearance of the adult-type $K^+$ currents ($I_{K,f}$ and $I_{K,n}$), such that by P59 $Myo7a^{fl/fl}Myo15$-cre$^{+/-}$ IHCs exhibited an immature current profile (Fig. 5f–i; Supplementary Table 3), including a slowly-activating outward ($I_K$) and an inward rectifier ($I_{K1}$) $K^+$ current[14,16]. IHCs from control mice ($Myo7a^{fl/fl}$: Fig. 5f–h; Supplementary Table 3; $Myo15$-cre$^{+/-}$: Supplementary Fig. 5; Supplementary Table 4) had the characteristic current profile of mature wild-type mice[9,14]. The size of IHCs, which is reported as cell membrane capacitance ($C_m$: Fig. 5i), became smaller with age in both $Myo7a^{fl/fl}$ and $Myo7a^{fl/fl}Myo15$-cre$^{+/-}$ IHCs, although it occurred earlier in the latter. The smaller $C_m$ is comparable to that measured in early postnatal IHCs[14]. The resting membrane potential ($V_m$) remained relatively constant as a function of age ($V_m$: Fig. 5i). This is because the main $K^+$ current contributing to the resting $V_m$ in IHCs is $I_{K,n}$ in adult $Myo7a^{fl/fl}$ mice and $I_{K1}$ in pre-hearing and "adult" $Myo7a^{fl/fl}Myo15$-cre$^{+/-}$ mice. The in vivo consequence of this progressive regression in IHC maturity was that while at P15 $Myo7a^{fl/fl}Myo15$-cre$^{+/-}$ mice had normal hearing thresholds (Fig. 4j), they were profoundly deaf at P62 (Supplementary Fig. 6). Adult $Myo7a^{fl/fl}Myo15$-cre$^{+/-}$ mice also exhibit the classical head bobbing and hyperactive circling behavior characteristic of $Myo7a$ mutant mice[32]. Both $Myo7a^{fl/fl}$ and $Myo15$-cre$^{+/-}$ mice showed normal hearing thresholds (Supplementary Fig. 6).

**The MET current ensures that IHCs retain adult $K^+$ currents.** To confirm the hypothesis that the resting MET current is required for IHCs to maintain their mature biophysical characteristics, we set up organotypic cultures from the adult cochlea. For these experiments we used gerbils (P18-P24), instead of mice, because their adult IHCs are more robust and can survive in vitro for the long periods required to test the hypothesis. Initially, cochleae were cultured for 1 to 8 days in the DMEM/F12 medium supplemented with $CaCl_2$ (final concentration: 1.3 mM $Ca^{2+}$, see Methods). Compared to IHCs recorded from acutely dissected gerbil cochleae (P21-P28: Fig. 6a, f), cells under the above culturing conditions gradually lost $I_{K,f}$, the most distinctive current expressed in post-hearing IHCs (Fig. 6b, c, f). By the 5$^{th}$ day in culture, $I_{K,f}$ was no longer present in IHCs while a large $I_K$ was evident (Fig. 6c, f). The main differences between the in vivo and our in vitro culture conditions, is that in the latter the IHC stereociliary bundles are not stimulated by sound, the endocochlear potential is absent and the resting open probability of the MET channel is very low because of the high extracellular $Ca^{2+}$ concentration (1.3 mM)[4]. Therefore, in vitro, only a very small or no depolarizing inward MET current is likely to be present[5,8]. By keeping IHCs depolarized using high $K^+$ in the culture medium (40 mM), as a proxy for the in vivo receptor potential caused by the resting MET current, $I_{K,f}$ was retained for longer periods (Fig. 6d, e, f). The direct contribution of the resting MET current in maintaining the adult IHC biophysical identity was tested by using a medium containing the expected in vivo mature endolymphatic $Ca^{2+}$ (40 µM), which substantially increases the resting MET current[4,27]. In 40 µM $Ca^{2+}$ (Fig. 6g, h, k), the size of $I_{K,f}$ was significantly higher compared to that measured in 1.3 mM $Ca^{2+}$ (Fig. 6b, c, k; $P = 0.006$, two-way ANOVA) or when 40 mM $K^+$ was used (Fig. 6d, e, f: $P = 0.0166$, two-way ANOVA). When cochlear cultures were incubated with 40 µM $Ca^{2+}$ together with 50 µM of the MET channel blocker d-tubocurarine[33,34] (Fig. 6i, j, l), which has been shown to be non-toxic to IHCs in long-lasting cochlear cultures[34], IHCs showed a strongly reduced $I_{K,f}$ (Fig. 6k) compared to when 40 µM $Ca^{2+}$ was used alone (Fig. 6k, $P = 0.0001$, two-way ANOVA), but produced similar results to when the open probability of the MET channel was very low (1.3 mM $Ca^{2+}$, black circles: Fig. 6k, $P = 0.6155$, two-way ANOVA). Blocking the MET channel caused IHCs not only to rapidly lose $I_{K,f}$ but also to express the inward rectifier $K^+$ current $I_{K1}$, which is normally present in pre-hearing cells[16] (Fig. 6l). $I_{K1}$ was already evident in 4-day cultured IHCs and became larger over time (253 ± 54 pA, $n = 13$, P18-P26 + 4–8 days). Although $I_{K1}$ was also seen in IHCs with partially closed MET channels (1.3 mM $Ca^{2+}$: 103 ± 22 pA, $n = 7$, P18 + 8 days) it was significantly smaller than that measured in the presence of 40 µM $Ca^{2+}$ and d-tubocurarine (P18 + 8 days: 393 ± 153 pA, $n = 4$, $P = 0.0324$, $t$-test).

**Normal efferent innervation requires the MET current.** Synaptic activity or exocytosis is known to change between pre- and post-hearing IHCs in terms of $Ca^{2+}$-efficiency[12]. Presynaptic activity was estimated by measuring the size of the $Ca^{2+}$ current ($I_{Ca}$) and the induced increase in cell membrane capacitance ($\Delta C_m$) following depolarizing voltage steps[12]. $I_{Ca}$ in IHCs from both young adult ($-191 ± 21$ pA, $n = 8$, 3 mice, P16-P17) and adult ($-158 ± 12$ pA, $n = 5$, 3 mice, P48) $Myo7a^{fl/fl}Myo15$-cre$^{+/-}$ mice was not significantly different ($P = 0.1252$, one-way ANOVA) to those measured in control $Myo7a^{fl/fl}$ IHCs ($I_{Ca}$: $-156 ± 13$ pA, $n = 5$, 2 mice, P16-P17; $-129 ± 5$ pA, $n = 4$, 3 mice, P49) (Fig. 7a–d). In addition, the corresponding $\Delta C_m$ was also similar between $Myo7a^{fl/fl}Myo15$-cre$^{+/-}$ (17.5 ± 1.9 fF, $n = 8$, P16-P17; 14.6 ± 1.0 fF, $n = 5$, P48) and control $Myo7a^{fl/fl}$ IHCs (17.4 ± 4.6 fF, $n = 5$, P16-P17; 13.6 ± 0.7 fF, $n = 4$, P49; $P = 0.652$, one-way ANOVA). The apparent larger inactivation of $I_{Ca}$ in

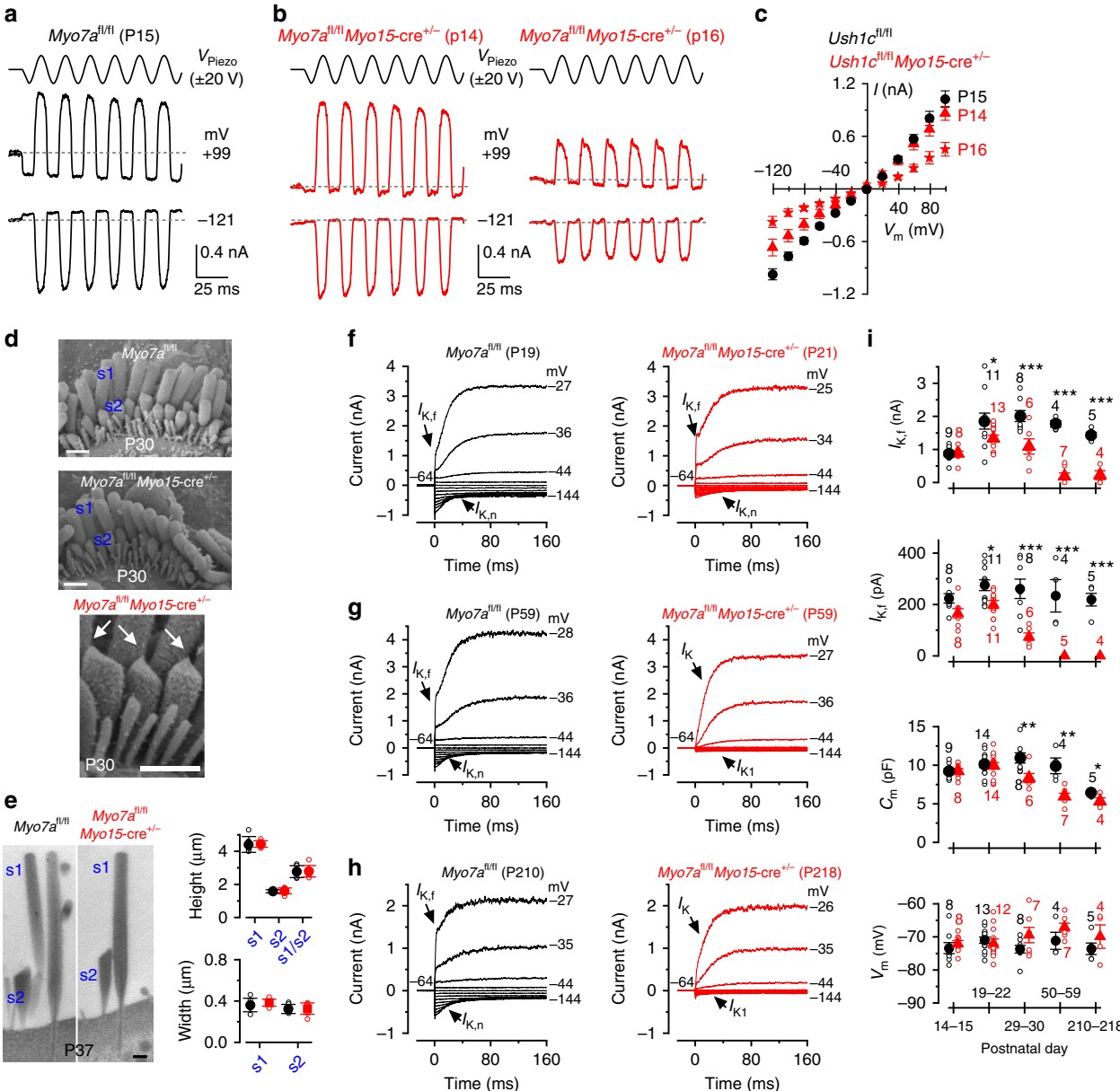

**Fig. 5** The functional MET current is required for retaining mature IHC characteristics. **a**, **b** MET currents recorded from mature IHCs of control $Myo7a^{fl/fl}$ (**a**, P15) and $Myo7a^{fl/fl}Myo15$-cre$^{+/-}$ (**b**, P14 and P16) mice. Protocol as in Fig. 2a. **c** Peak-to-peak MET current-voltage curves obtained from six P15 $Myo7a^{fl/fl}$ (2 mice), four P14 $Myo7a^{fl/fl}Myo15$-cre$^{+/-}$ and six P16 $Myo7a^{fl/fl}Myo15$-cre$^{+/-}$ IHCs (3 mice). **d** Scanning electron microscopy images showing the stereociliary bundle of apical-coil IHC from P30 control $Myo7a^{fl/fl}$ and $Myo7a^{fl/fl}Myo15$-cre$^{+/-}$ mice. s1 and s2 represent the tallest and the second tallest stereocilia, respectively. The arrows in the middle panel highlights that tip links, which are required to gate the MET current, are present in $Myo7a^{fl/fl}Myo15$-cre$^{+/-}$ IHCs. Scale bars, 1 µm. **e** Transmission electron microscopy images (left) showing the stereocilia s1 and s2 (see panel **d**) of apical-coil IHCs from P37 control $Myo7a^{fl/fl}$ and $Myo7a^{fl/fl}Myo15$-cre$^{+/-}$ mice. Scale bars, 400 nm. The right panels show that the height and width of s1 and s2 stereocilia in IHCs are comparable between the two genotypes ($Myo7a^{fl/fl}$ s1: height 4.42 ± 0.48 µm, width 0.36 ± 0.07 µm, $n = 6$ IHCs, s2: height 1.58 ± 0.10 µm, width 0.32 ± 0.04 µm, $n = 7$, s1/s2: 2.77 ± 0.36, $n = 6$; $Myo7a^{fl/fl}Myo15$-cre$^{+/-}$ s1: 4.44 ± 0.20 µm, $P < 0.8915$, 0.38 ± 0.03 µm, $P < 0.4441$, $n = 7$, s2: 1.61 ± 0.18 µm, $P < 0.7849$, 0.33 ± 0.06 µm, $P < 0.8959$, $n = 7$, s1/s2: 2.80 ± 0.35, $P < 0.9059$, $n = 6$). **f–h** Potassium currents recorded from IHCs of young adult (**f**, P19 and P21), adult (**g**, P59) and aged (**h**, P210 and P218) control $Myo7a^{fl/fl}$ (left panels) and $Myo7a^{fl/fl}Myo15$-cre$^{+/-}$ (right panels) mice. Protocols are as in Fig. 3a. **i** Size of $I_{K,f}$, $I_{K,n}$, membrane capacitance ($C_m$) and resting membrane potential ($V_m$) as a function of postnatal age. Number of IHC investigated is shown above or below the symbols (for more details see Supplementary Table 3). Statistical tests are performed with two-way ANOVA with Bonferroni correction. Average data is shown as mean ± SEM (**c**, **i**) or mean ± SD (**e**)

$Myo7a^{fl/fl}Myo15$-cre$^{+/-}$ IHCs (Fig. 7c) was due to the activation of the SK2 current (see also Fig. 7f), which is characteristic of pre-hearing IHCs[35,36].

In the adult mammalian cochlea, the efferent fibers either form axo-somatic synapses with OHCs or axo-dendritic synapses with the afferent fibers that contact IHCs[37]. IHCs are directly

innervated by cholinergic efferent endings only during early stages of development, but no longer respond to acetylcholine (ACh) from about P18 onwards[35,36,38]. Adult IHCs from P22 $Myo7a^{fl/fl}$ and $Myo7a^{fl/fl}Myo15$-cre$^{+/-}$ mice did not respond to extracellular ACh (Fig. 7g; Supplementary Fig. 7a, b) and no longer showed axo-somatic efferent contacts at P20

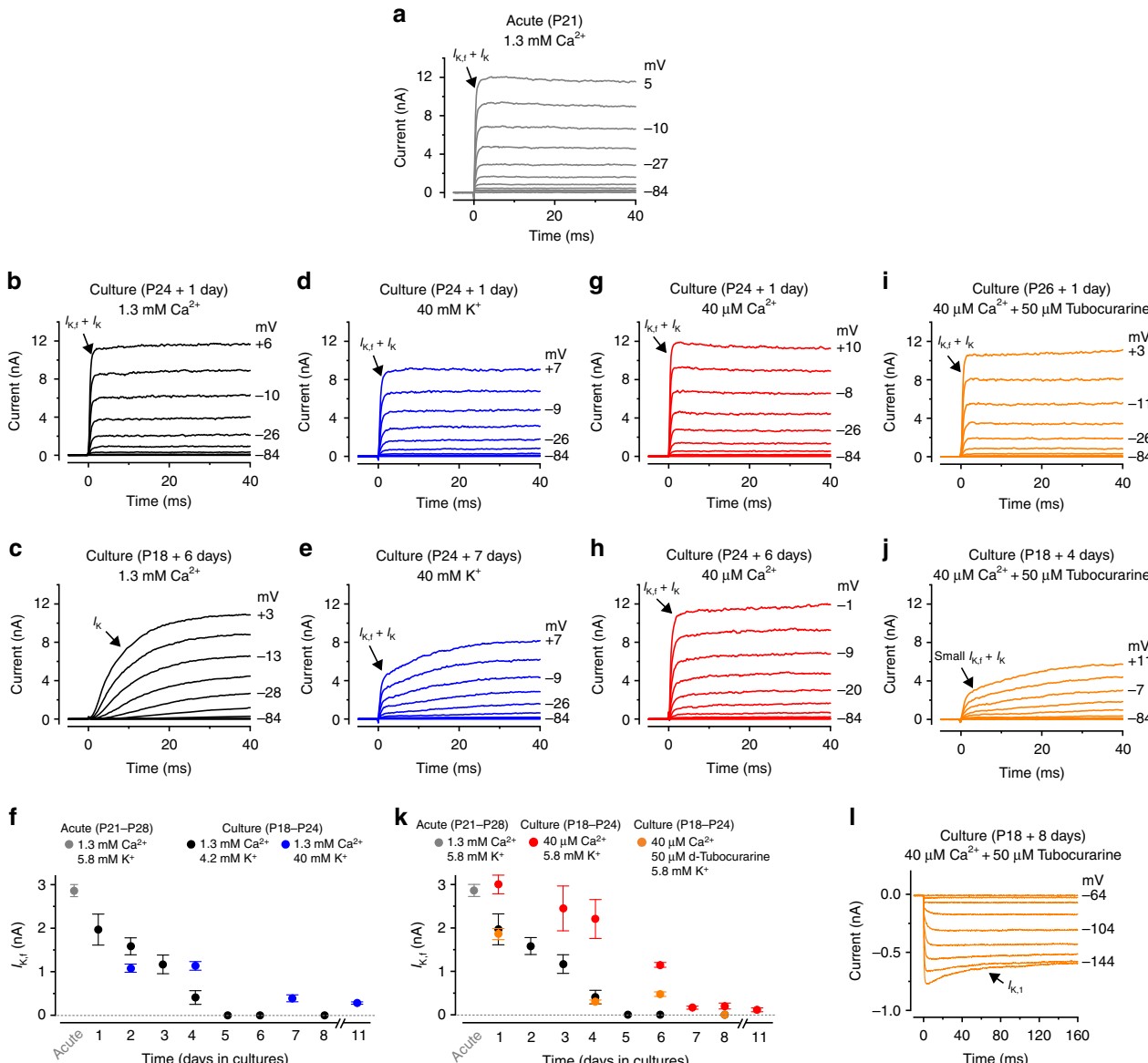

**Fig. 6** The resting MET current drives the maintenance of mature IHC K$^+$ currents in cochlear cultures. **a** Potassium outward currents recorded from a P21 IHC of an acutely dissected gerbil cochlea. Recordings were performed using a perilymph-like solution (1.3 mM extracellular Ca$^{2+}$; 5.8 mM K$^+$). Currents were elicited using 10 mV depolarizing voltage steps from $-84$ mV. Note the fast activating $I_{K,f}$ characteristic of adult IHCs (arrow). **b, c** Currents recorded from IHCs maintained in cochlear culture for 1 day (**b**) or 6 days (**c**). The culture medium bathing the cochlea (see Methods) contained 1.3 mM Ca$^{2+}$ and 4.2 mM K$^+$. **d, e** Currents recorded from IHCs maintained in culture for 2 day (**d**) or 7 days (**e**). Culture medium contained 1.3 mM Ca$^{2+}$ and 40 mM K$^+$. **f** Size of $I_{K,f}$ recorded from IHCs of acutely isolated gerbil cochleae (gray circle, $n = 6$, P21-P28, 3 gerbils) or maintained in culture. Culture conditions were: (1) normal medium (black circles, number of IHCs from left to right: 7,3,4,10,2,6,7; from 9 cochlear cultures; gerbil age at the day of culturing: P18-P24); 2) medium supplemented to 40 mM K$^+$ (blue circles, number of IHCs: 6,6,5,7, from 4 cochlear cultures, P18-P24). **g, h** Currents recorded from IHCs maintained in culture for 1 day (**g**) or 6 days (**h**) with a medium containing 40 μM Ca$^{2+}$. **i, j, l** Currents from IHCs maintained in cochlear culture for 1 day (**i**), 4 days (**j**) or 8 days (**l**) with a medium containing 40 μM Ca$^{2+}$ and 50 μM of the MET channel blocker d-tubocurarine. The inward rectifier K$^+$ current ($I_{K1}$) **l** was obtained using 10 mV hyperpolarizing voltage steps from $-64$ mV. **k** Size of $I_{K,f}$ recorded from IHCs of acutely dissected cochleae (gray circle: same as in panel **f**), maintained in culture with a normal medium (black circles: as in panel **f**), a medium containing 40 μM Ca$^{2+}$ (red circles: number of IHCs: 5,10,9,5,3,5,8, from 9 cochlear cultures, P18-P24) and a medium containing 40 μM Ca$^{2+}$ and 50 μM d-tubocurarine (orange circles: 4,6,6,4, from 4 cochlear cultures, P18-P24). Average data is shown as mean ± SEM

(Supplementary Fig. 7c, d), indicating that the re-wiring of maturing IHCs occurred normally. However, at older ages (P33-P49), while IHCs from control $Myo7a^{fl/fl}$ mice remained insensitive to ACh (Fig. 7e), those from $Myo7a^{fl/fl}Myo15$-cre$^{+/-}$ mice responded with an instantaneous current (Fig. 7f) that increased the slope conductance at $-84$ mV ($P = 0.0045$, t- test, Fig. 7g). In pre-hearing IHCs the ACh-dependent current is mediated by Ca$^{2+}$ entering through α9α10nAChRs, which then activates SK2 channels[35,36]. Both currents components were observed when ACh was applied to $Myo7a^{fl/fl}Myo15$-cre$^{+/-}$IHCs (Fig. 7f), with the outward current mainly carried by SK2 channels and the inward relaxing current most likely due to cations entering the IHC through α9α10nAChRs from the holding potential of $-84$ mM[35].

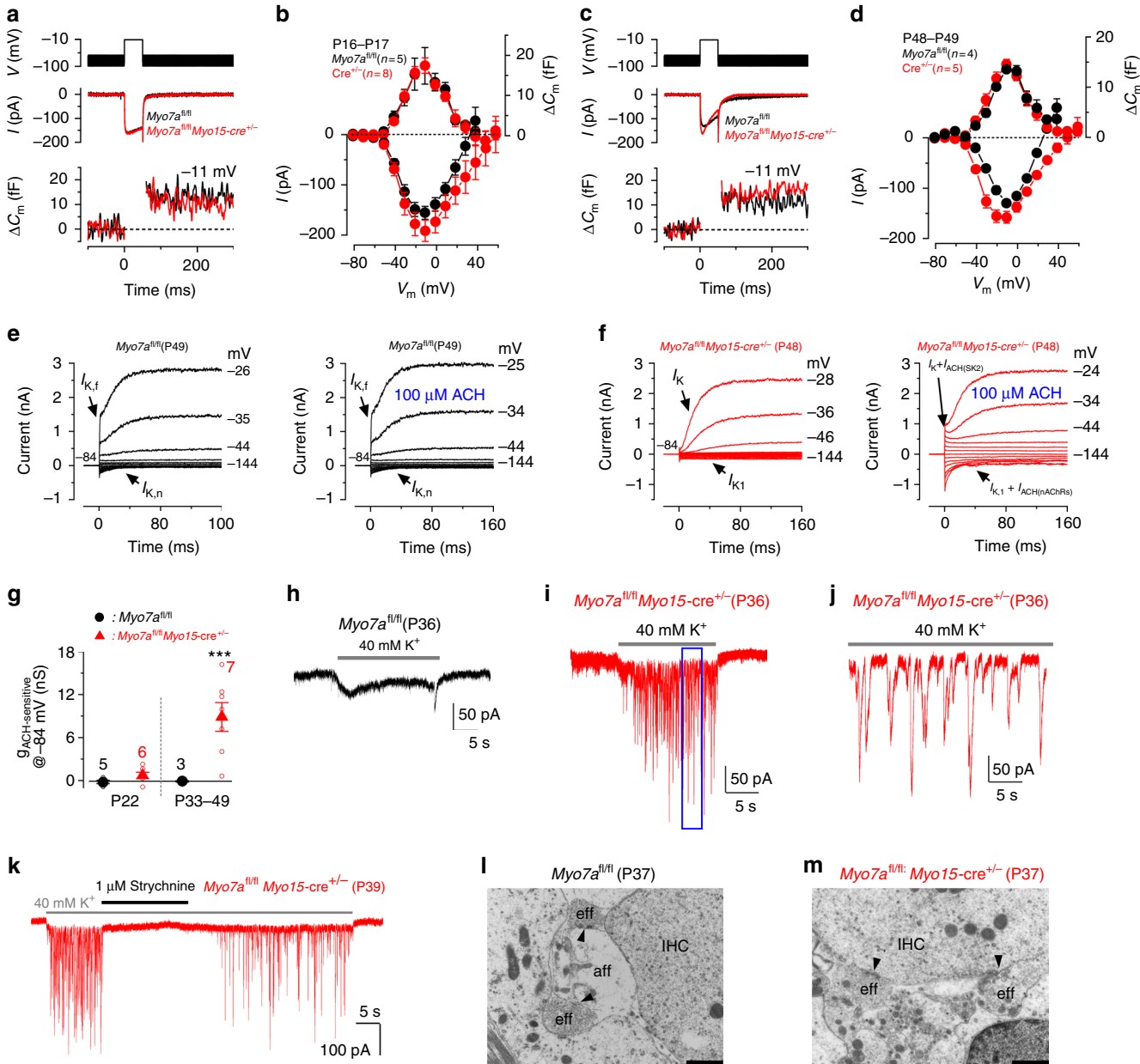

**Fig. 7** IHC exocytosis is unaffected in adult $Myo7a^{fl/fl}Myo15$-cre$^{+/-}$ mice but efferent input is altered. **a−d** Calcium current ($I_{Ca}$) and corresponding changes in membrane capacitance ($\Delta C_m$) recorded from IHCs of P16-P17 (**a, b**) and P48-P49 (**c, d**) control $Myo7a^{fl/fl}$ (black) and knockout $Myo7a^{fl/fl}Myo15$-cre$^{+/-}$ mice (red). Recordings were obtained in response to 50 ms voltage steps (10 mV increments) from −81 mV and using 1.3 mM extracellular Ca$^{2+}$ and at body temperature. For clarity, only maximal responses at −11 mV are shown. Panels **b** and **d** shows average peak $I_{Ca}$ (bottom) and $\Delta C_m$ (top) curve from control $Myo7a^{fl/fl}$ and $Myo7a^{fl/fl}Myo15$-cre$^{+/-}$ IHCs. **e, f** Membrane currents recorded from adult IHCs of control $Myo7a^{fl/fl}$ (**e**) and $Myo7a^{fl/fl}Myo15$-cre$^{+/-}$ (**f**) mice before (left panels) and during superfusion of ACh (right panels). ACh elicited an instantaneous current only in the $Myo7a^{fl/fl}Myo15$-cre$^{+/-}$ IHC. **g** Steady-state slope conductance of the ACh-sensitive current ($g_{ACh-sensitive}$) at different age ranges in control $Myo7a^{fl/fl}$ (black symbols) and $Myo7a^{fl/fl}Myo15$-cre$^{+/-}$ (red symbols) IHCs. The isolated $g_{ACh-sensitive}$ was obtained by subtracting the control current (left panels in **e, f**) from that in the presence of 100 μM ACh (right panels)[35]. Number of IHCs is shown near the symbols (2 $Myo7a^{fl/fl}$ mice at P22 and P33-P49; 3 $Myo7a^{fl/fl}Myo15$-cre$^{+/-}$ mice at P22; 5 $Myo7a^{fl/fl}Myo15$-cre$^{+/-}$mice at P34-P48). **h–k** Voltage-clamp recordings obtained from adult IHCs of control $Myo7a^{fl/fl}$ (**h**, P36) and $Myo7a^{fl/fl}Myo15$-cre$^{+/-}$ (**i, j**, P36; **k** P39) mice during the superfusion of 40 mM extracellular K$^+$. Synaptic currents were only evoked in $Myo7a^{fl/fl}Myo15$-cre$^{+/-}$ IHCs. Panel **j** shows an expanded time scale of the blue area shown in panel (**i**). Panel **k** shows the effect of 1 μM strychnine on the ACh-induced synaptic currents. **l, m** TEM showing the IHC synaptic region from P37 control $Myo7a^{fl/fl}$ (**l**) and $Myo7a^{fl/fl}Myo15$-cre$^{+/-}$ (**m**) mice. IHCs from $Myo7a^{fl/fl}$ mice showed the characteristic synaptic organization of mature cells with efferent terminals forming axo-dendritic contact with the afferent fibers, as seen by the presence of active zones (**l**, black arrows). Scale bars, 1 μm. In $Myo7a^{fl/fl}Myo15$-cre$^{+/-}$ mice efferent fibers make direct axo-somatic contact with the IHC (**m**), which is characteristic of pre-hearing IHCs. Average data is shown as mean ± SEM

In order to establish whether re-appearance of ACh receptors in adult $Myo7a^{fl/fl}Myo15$-cre$^{+/-}$ IHCs was associated with re-establishment of axosomatic efferent synapses, we perfused the cochlea with a solution containing 40 mM KCl. High K$^+$ is expected to depolarize the efferent synaptic terminals and trigger the release of ACh-containing vesicles, which generate synaptic currents in IHCs superimposed upon a sustained inward current due to the direct effect of KCl on the cell's basolateral membrane[38,39]. Synaptic inward currents were only seen in $Myo7a^{fl/fl}Myo15$-cre$^{+/-}$ IHCs (Fig. 7h–j),with a frequency (5.7 ± 0.7 Hz, 715 events, $n = 7$, 4 mice) and amplitude (98 ± 19 pA, 594 events, $n = 7$) not significantly different from those previously reported in pre-hearing IHCs ($P = 0.68$ and $P = 0.107$, $t$-test, respectively)[39]. Reversible block of the ACh-mediated currents by 1 µM strychnine (Fig. 7k), a potent blocker of $\alpha9\alpha10$nAChRs[40], confirmed the re-expression of these receptors in adult $Myo7a^{fl/fl}Myo15$-cre$^{+/-}$ IHCs. TEM experiments on P37 mice ($\geq 3$ mice for each genotype) indicated that while IHCs from $Myo7a^{fl/fl}$ mice showed the characteristic axo-dendritic contacts between efferent and afferent fibers (Fig. 7l), $Myo7a^{fl/fl}Myo15$-cre$^{+/-}$mice underwent major re-wiring with efferents re-forming direct axo-somatic innervation with adult IHCs (Fig. 7m).

## Discussion

Here we show that the MET current is essential to establish and maintain the key biophysical and morphological properties characteristic of adult IHCs. During the second post-natal week in mice, which is just before the onset of hearing at P12, open MET channels keep the IHC membrane potential at the threshold for Ca$^{2+}$ action potentials (APs), promoting their functional differentiation into mature graded sensory receptors able to encode sound stimuli. We also found that adult IHCs require functional MET channels to maintain their adult profile. The absence of the MET current during adult stages caused IHCs to lose their identity, regressing into an immature-like biophysical and morphological profile. Adult IHCs without functional MET channels became re-innervated by inhibitory cholinergic efferent neurons, which normally make transient axo-somatic contacts only with pre-hearing IHCs.

Pre-hearing IHCs of altricial rodents exhibit spontaneous Ca$^{2+}$-dependent APs[5,10,14] that can be modulated by the inhibitory efferent neurotransmitter ACh[38] and by excitatory spontaneous ATP-induced Ca$^{2+}$-waves in non-sensory cells, which synchronize the activity of several IHCs[10,28,41]. Indeed, patterned bursts of APs have been recorded from auditory afferent neurons prior to the onset of hearing in mice[42,43], and both their survival[11] and the refinement of tonotopic maps in the auditory brainstem[13,44] rely on the firing activity of pre-hearing IHCs. However, the refinement of the auditory pathway mainly occurs during the first postnatal week[11,44,45], a finding supported by evidence that surgical removal of the cochlea causes a substantial loss of cochlear nucleus neurons mainly during this period[46,47]. Despite the crucial role of APs in developing IHCs, spiking activity is not suitable to detect sound because it would modulate and degrade auditory stimuli in the mature cochlea. Therefore, just before hearing onset immature IHCs undergo major changes involving the acquisition and/or elimination of different ion channels and other membrane proteins[9,12,14]. Intracellular Ca$^{2+}$ transients associated with a precise frequency and/or pattern of AP activity can activate transcription factors controlling the expression of proteins linked to development[48,49]. In cochlear IHCs, repetitive Ca$^{2+}$ APs during the second postnatal week are driven by a standing depolarizing MET current[5], resulting from the progressive reduction in endolymphatic Ca$^{2+}$ concentration[5] that, by reducing adaptation, increases the MET channel open

probability. Therefore, AP activity driven by the greater standing MET current during the few days preceding hearing onset could provide the cue or trigger for the last step of IHC development by influencing gene expression. We demonstrate that genetic manipulations that abolish the contribution of the MET current in vivo during the second postnatal week prevented IHCs from becoming mature auditory receptors. We propose that, during pre-hearing stages of development, the primary function of IHC AP activity could switch from refining the auditory circuitry[13] to promoting the maturation of the sensory receptor itself.

The normal AP activity in IHCs from pre-hearing $Myo7a^{fl/fl}Myo15$-cre$^{+/-}$ mice meant that their auditory system was able, at least initially, to mature fully as judged by the presence of normal ABRs (P15), absence of axo-somatic efferent terminals on P20-P22 IHCs, and a normal IHC basolateral current profile at P20. However, the loss of IHC MET current after the onset of hearing caused a progressive down-regulation of the adult-type basolateral ion channels carrying $I_{K,f}$ and $I_{K,n}$. By two months of age, IHCs from $Myo7a^{fl/fl}Myo15$-cre$^{+/-}$ mice have the same size and basolateral membrane currents as pre-hearing cells, including the re-expression of $I_{K1}$[16] and $I_{SK2}$[35,36]. A similar progressive loss of $I_{K,f}$ was also observed in cultured cochlear IHCs, in which cells are not stimulated by sound and the resting open probability of the MET current is either reduced or blocked. In vitro experimental manipulations that increased the MET channel open probability (endolymphatic-like Ca$^{2+}$ concentration) or mimicked the receptor potential due to sound stimulation (IHC depolarization) were able to maintain the adult current profile of cultured IHCs for longer. Calcium entering via the MET channel has been shown to be required for maintaining the morphological stability of transducing stereocilia in early postnatal hair cells[50]. Although in the presence of mature endolymphatic-like solution Ca$^{2+}$ entering through the MET channel seems to be cleared from the stereocilia by the plasma membrane Ca$^{2+}$ ATPase type 2 (PMCA2), some of it could diffuse into the cell body[51] and, together with the depolarization by the MET current, potentially regulate gene expression and as such the maintenance of the adult IHC current profile. The only adult-like function that was not affected by the loss of the MET channel was that associated with the synaptic machinery, indicating that once it has matured, a process that is controlled by a critical period of pre-hearing APs[12], it becomes independent of IHC activity. Loss of ribbon synapses in the adult cochlea occurs following acoustic trauma or ototoxic antibiotics[52], conditions that cause either a temporary or permanent IHC damage. IHCs from aging mice have also been shown to gradually lose ribbon synapses[53]. The fact that exocytosis in IHCs from $Myo7a^{fl/fl}Myo15$-cre$^{+/-}$ mice was normal at 7 weeks, despite IHCs starting to become smaller, may be due to ribbon loss being a slow process[53].

While auditory ribbon synapses appeared to be resilient to post-maturational changes in MET function, the efferent fibers were more plastic. The efferent endings make transient cholinergic axo-somatic contacts with immature IHCs[54,55]. After the onset of hearing, efferent terminals only form axo-dendritic synapses with the auditory afferent fibers contacting IHCs[56] and expression of the postsynaptic $\alpha9$nAChRs and SK2 channels is rapidly down-regulated[54]. Therefore, adult IHCs lose the ability to respond to ACh[38,54]. However, it has been shown that efferent endings can re-innervate IHCs following cochlear damage[57] or in aging mice[53,58]. We found that the re-emergence of axo-somatic efferent innervation with IHCs can also occur in mature mice with no functional MET current.

The above findings, including the reduced size of the IHCs, the direct re-innervation by efferent cholinergic fibers, the complete down-regulation of the adult-like basolateral membrane ion channels and the ability to fire APs, indicate that functional MET

channels are required for mature IHCs to maintain their biophysical and morphological identity. Without functional MET channels, IHCs revert to an earlier, pre-hearing stage of development. It would be interesting in the future to determine whether the re-establishment of the immature-like biophysical and morphological profile in IHCs without working MET channels is the first stage of an intrinsic attempt to repair the faulty cochlea by "recapitulating" early development. In any case, this functional regression shows a surprising plasticity in the mature cochlea in response to events that abolish normal MET currents, which could occur due to either noise[59,60], aging[61] or in forms of progressive hearing loss, such as Usher syndrome type III[62].

## Methods

**Ethics statement.** All animal studies were performed in the UK and licensed by the Home Office under the Animals (Scientific Procedures) Act 1986 and were approved by the University of Sheffield and University of Sussex Ethical Review Committees. ABR measurements (see below) were conducted under anesthesia using ketamine (100 mg/Kg, Fort Dodge Animal Health) and xylazine (10 mg/Kg, Rompun 2%, Bayer), which were administered with intraperitoneal injection as previously described[63]. At the end of the experiments, which lasted up to 40 minutes, mice were usually culled by a schedule 1 method. In a few animals, ABR measurements were followed by the recovery of the mice from anesthesia with intraperitoneal of atipamezole mix[63].

**Animal strains.** For the conditional knockout mice, the targeted *tm1a* allele for both *Myo7a* (Myo7a$^{tm1a(EUCOMM)Wtsi}$ allele ID: 4431921) and *Ush1C* (Ush1c$^{tm1a(KOMP)Wtsi}$ allele ID: 4363497) were generated by the Mouse Genetics Programme at the Wellcome Trust Sanger Institute (Cambridge, UK). For *Myo7a* the critical exons 10 and 11 were floxed, while for *Ush1c* exons 5–8 were floxed. The *tm1c* allele was obtained by crossing the *tm1a* mouse to a FLPeR carrying mouse (Rosa26$^{Fki}$). The *tm1d* alleles, which were used for the experiments, were obtained by crossing the *tm1c* mouse (Myo7a$^{fl/fl}$ and Ush1c$^{fl/fl}$) with the Myo15-cre mice and genotyped as previously described[24]. The genotyping protocols for mutant mice were also previously described: *Myo7a* (Myo7a$^{sh6J/sh6J}$)[31] and *Pcdh15* (Pcdh15$^{av3J}$)[15]. For the conditional knockout mice (*tm1c* alleles) genomic DNA was amplified under standard reaction conditions using Promega Taq DNA polymerase (Invitrogen, UK). The reaction mix was incubated at 94 °C for 5 min followed by 40 cycles of 94 °C for 30 s, 58 °C for 45 s and 72 °C for 90 s, followed by 72 °C for 5 min. Primer pairs and product size are as follows: *Myo7a*: forward, 5′-GGGGAGAGAAAGCGGGTGTG- 3′ and reverse, 5′- AAGCTGGACTCTCTGGT GGC-3′, producing a 360-bp product for the WT and a 460-bp product for the *tm1c* homozygous mutant; *Ush1C* : forward, 5′-ACAGAGCCGTGGGTTCATTC-3′ and reverse, 5′-GTAATGGAGCTGAGGCAGGG-3′, producing a 326-bp product for the WT and a 430-bp product for the *tm1c* homozygous mutant.

**Tissue preparation.** Inner hair cells (IHCs) were studied in acutely dissected organs of Corti from postnatal day 7 (P7) to P218 mice and P21-P28 gerbils, where the day of birth is P0. In mice, experiments were performed on IHCs positioned in the apical coil of the cochlea (around 12 kHz)[64]. Gerbil IHCs were positioned at a frequency range of 250–420 Hz[65]. Animals of either sex were killed by cervical dislocation and the organ of Corti dissected in extracellular solution composed of (in mM): 135 NaCl, 5.8 KCl, 1.3 CaCl₂, 0.9 MgCl₂, 0.7 NaH₂PO₄, 5.6 D-glucose, 10 Hepes-NaOH. Sodium pyruvate (2 mM), amino acids and vitamins were added from concentrates (Fisher). The pH was adjusted to 7.5 (osmolality ~308 mmol kg$^{-1}$). The dissected apical coil of the organ of Corti was transferred to a microscope chamber, immobilized using a nylon mesh fixed to a stainless steel ring and viewed using an upright microscope (Leica DM LFS, Germany or Nikon FN1). Hair cells were observed with Nomarski differential interface contrast optics (×63 water immersion objectives). To expose the basolateral surface of the cells, a small tear was made in the epithelium with a suction pipette (tip diameter 3–4 μm) filled with extracellular solution.

**Single-cell electrophysiology.** Membrane currents and voltage responses were investigated under whole cell voltage or current clamp, respectively, either at room temperature (20–25 °C) or body temperature (34–37 °C), using Optopatch amplifiers (Cairn Research Ltd, UK). Patch pipettes, with resistances of 2–3 MΩ, were pulled from soda glass capillaries (Harvard Apparatus Ltd, Edenbridge, UK). The shank of the electrode was coated with surf wax (Mr Zoggs Sex Wax, CA, USA) to reduce the electrode capacitative transient. When investigating the basolateral properties of IHCs, the intracellular solution was composed of (mm): 131 KCl, 3 MgCl₂, 1 EGTA-KOH, 5 Na₂ATP, 5 Hepes-KOH, 10 Na-phosphocreatine (pH 7.28, 294 mmol kg$^{-1}$). When investigating MET currents or membrane capacitance, a Cs-glutamate intracellular solution was used composed of (mM): 106 Cs-glutamate, 20 CsCl, 3 MgCl₂, 1 EGTA-CsOH, 5 Na₂ATP, 0.3 Na₂GTP, 5 Hepes-CsOH, 10 Na₂-phosphocreatine (pH 7.3, 294 mmol kg$^{-1}$). Data acquisition was

performed using a Digidata 1440 A with pClamp software 10.0 (Axon Instruments, CA, USA) or a CED Power 1401 with Signal Software (CED, Cambridge, UK). Data were filtered at 2.5 kHz (8-pole Bessel) and sampled at ≥ 5 kHz. Origin software (OriginLab, Northampton, MA, USA) was used to perform offline data analysis. Membrane potentials were corrected for the residual series resistance $R_s$ after compensation (usually 80%) and liquid junction potential (LJP), which for Cs-glutamate and KCl intracellular solutions were −11 mV and −4 mV, respectively.

When investigating basolateral membrane properties, the size of $I_{K,f}$ was measured near –25 mV and at either 0.7 ms or 1.0 ms after the start of the voltage step, while $I_{K,n}$ was measured as the difference between the peak and steady state of the deactivating inward current at −124 mV. Steady-state total currents were measured at 160 ms, at a potential of 0 mV (extrapolated from the current-voltage curves). The holding potentials used for these recordings were set at either −84 mV or −64 mV (normally specified in the figure legends).

The presence of inhibitory postsynaptic currents in IHCs, which are caused by the release of ACh-containing vesicles from the efferent terminals, was obtained by superfusing 40 mM extracellular K⁺ in order to depolarize the efferent fibers. Event analysis was performed using the Mini Analysis Program (Synaptosoft Inc., USA). Note that the number of events used to calculate the frequency of inhibitory postsynaptic currents (715 events) was larger than that used for the amplitude (594 events: see Results linked to Fig. 7i–l) since the presence of partially overlapping events prevented the accurate assessment of the baseline.

Real-time changes in membrane capacitance ($\Delta C_m$) were performed at body temperature as previously described[12]. Briefly, a 4 kHz sine wave of 13 mV RMS was applied to IHCs from −81 mV, unless otherwise stated, and was interrupted for the duration of the voltage step. The capacitance signal from the Optopatch was filtered at 250 Hz and sampled at 5 kHz. $\Delta C_m$ was measured by averaging the $C_m$ trace over a 200 ms period following the voltage step and subtracting the pre-pulse baseline. Data were acquired using pClamp software and a Digidata 1440 A (Molecular Devices). $\Delta C_m$ experiments were performed in the presence of 30 mM TEA and 15 mM 4-AP (Fluka) to block the BK current ($I_{K,f}$)[9] and delayed rectifier K⁺ currents ($I_K$, previously called $I_{K,neo}$ in the case of pre-hearing IHCs and $I_{K,s}$ in mature IHCs), and linopirdine (80 μM: Tocris) to block $I_{K,n}$[14].

For mechanoelectrical transducer (MET) current recordings, a fluid jet driven by a piezoelectric disc was used to elicit bundle displacement as previously described[4]. The tip of the fluid jet pipette had a diameter of 8–10 μm and was positioned at about 8 μm from the bundles. Saturating mechanical stimuli were applied using 50 Hz sinusoids (filtered at 1 kHz, 8-pole Bessel) with a driver voltage ($V_{Piezo}$) to the fluid jet of ± 20 V, with positive deflections moving the stereocilia in the excitatory direction. For some experiments a gravity fed local perfusion system was used to extracellularly apply endolymph-like solution (300 μM Ca²⁺)[5] either alone or with 200 μM dihydrostreptomycin (DHS). The resting MET current becomes evident when the MET channels shut off in the inhibitory phase of the stimulus: the holding current minus the current present during inhibitory bundle deflection[4].

Unless otherwise stated, electrophysiological data was analyzed with PCLAMP software (Molecular Devices) and ORIGIN (Origin Lab, USA).

**Auditory brainstem responses.** Auditory brainstem responses (ABRs) were recorded from either male or female knockout mice and their littermate controls between P15 and P62. Recordings were performed in a soundproof chamber (MAC-3 Acoustic Chamber, IAC Acoustic, UK) as previously described[63]. Briefly, stimuli were delivered to the ear by calibrated loudspeakers (MF1-S Multi Field Speaker, Tucker-Davis Technologies, USA) placed 10 cm from the animal's pinna. Sound pressure was calibrated with a low-noise microphone probe system (ER10B + , Etymotic, USA). Experiments were performed using BioSigRZ software driving an RZ6 auditory processor (Tucker-Davis Technologies). Response thresholds were estimated from the resulting ABR waveform and defined as the lowest sound level where any recognizable feature of the waveform was visible. Responses were measured for click and pure tone stimuli of frequencies between 3.0 and 24 kHz. Stimulus sound pressure levels were typically 0–95 dB SPL, presented in steps of 5 dB. The brainstem response signal was averaged over 256 repetitions. Tone bursts were 5 ms in duration with a 1 ms on/off ramp time, which was presented at a rate of 42.6/sec.

**Two-photon confocal Ca²⁺ imaging.** For calcium dye loading, acutely dissected cochleae were incubated for 40 min at 37 °C in DMEM/F12, supplemented with fluo-4 AM (final concentration 10 μM; Thermo Fisher Scientific) as recently described[66]. Ca²⁺ signals were recorded using a two-photon laser-scanning microscope (Bergamo II System B232, Thorlabs Inc., USA) based on a mode-locked laser system operating at 800 nm, 80-MHz pulse repetition rate, < 100-fs pulse width (Mai Tai HP DeepSee, Spectra-Physics, USA). Images were captured with a 60x objective (LUMFLN60XW, Olympus, Japan) using a GaAsp PMT (Hamamatsu) coupled with a 525/40 bandpass filter (FF02-525/40-25, Semrock). Images were analyzed off-line using custom built software routines written in Python (Python 2.7, Python Software Foundation) and ImageJ (NIH). Ca²⁺ signals were measured as relative changes of fluorescence emission intensity ($\Delta F/F_0$).

**Cochlear culture preparation**. Cochlear cultures from P18-P24 gerbils were prepared as described previously[1]. Briefly, cochleae were dissected in HEPES buffered (10 mM, pH 7.2) Hanks' balanced salt solution (HBHBSS), placed onto collagen-coated glass coverslips, fed with 100–150 μl of medium containing 98% or 93% standard Dulbecco's modified Eagle's DMEM/F12 with additional 10 mM HEPES buffer, 2% or 7% foetal bovine serum (Biosera, UK) and 10 μg/ml ampicillin (Sigma) and maintained at 37 °C for up to 11 days. The standard DMEM/F12 solution containing 4.18 mM KCl and 1.09 mM $CaCl_2$ (Sigma, UK) was used, additionally supplemented with KCl or $CaCl_2$ depending on the experiments (see Results and Fig. 6). For the experiments designed to test the effect of the adult endolymphatic-like solution (20–40 μM $Ca^{2+}$)[6,67] the above medium was buffered with 2.2 mM HEDTA. For the recordings aimed at testing the effect of high extracellular $K^+$, the above medium was supplemented with 1 M KCl to reach the final desired concentrations (15 mM or 40 mM KCl). In order to block the MET channels in the presence of the endolymphatic-like solution (40 μM $Ca^{2+}$: see above), we used 50 μM of the MET channel blocker d-tubocurarine (Tocris, UK).

**Immunofluorescence microscopy**. Dissected inner ears from $Myo7a^{fl/fl}$ and $Myo7a^{fl/fl}Myo15$-cre$^{+/-}$ mice ($n = 5$ and $n = 8$ mice, respectively) were perfused with 4% paraformaldehyde in phosphate-buffered saline (PBS) for 45 min at 4 °C. Cochleas were microdissected, rinsed three times for 10 min, and incubated for 1 h at room temperature in PBS supplemented with 20% normal horse serum and 0.3% Triton X-100. The samples were then incubated overnight with the primary antibody: rabbit anti-myosin7a[68] (1/500) in PBS supplemented with 1% horse serum. The samples were rinsed three times for 10 min in PBS, and then incubated for 1 h with ATTO-488-conjugated goat anti-rabbit IgG antibody (Sigma-Aldrich, #18772, 1:500 dilution). Actin was labeled with ATTO-647N–conjugated phalloidin (Sigma-Aldrich, #65906, 1:200 dilution). Samples were then mounted in Fluorsave (Calbiochem, USA). The z-stack images were captured with a Zeiss LSM-700 confocal microscope equipped with a Plan Apochromat 63X/NA 1.4 oil immersion lens (Carl Zeiss, Jena, Germany).

**Scanning electron microscopy (SEM)**. The auditory bullae were isolated from the heads of the mice and opened to expose the cochleae which were fixed, by intra-labyrinthine perfusion using a fine hypodermic needle through the round window. The fixative was 2.5% glutaraldehyde in 0.1 M cacodylate buffer containing 3 mM calcium chloride (pH 7.3). Following perfusion the intact cochleae were immersed in the fixative for 2 h and then decalcified by immersion in 4.0% EDTA (pH 7.3) for 2 days at 4 °C. Subsequently the cochleae were dissected under cacodylate buffer to isolate approximately half turn segments of the organ of Corti from apex to base. These organ of Corti segments were post-fixed (2 h) in cacodylate buffered 1% $OsO_4$, before processing through the thiocarbohydrazide-Os-repeated procedure[69], followed by dehydration in an ethanol series and critical point drying. The organ of Corti samples were mounted on SEM support stubs with silver paint and sputter coated with a thin (ca 5 nm) layer of platinum before examination in a JEOL 6700 cold field-emission SEM operating at 3 or 5 kV. Images are a sample from 3 or more mice for each genotype.

**Transmission electron microscopy (TEM)**. For TEM cochleae were fixed as for SEM but postfixed by immersion for 1 h in 1% osmium tetroxide in 0.1 M cacodylate buffer, dehydrated and embedded in Spurr resin[70]. For ultrastructural evaluation of the nerve terminal region of IHCs, ultrathin (80–90 nm) sections were cut in radial planes from the apical coil, mounted on 200 mesh thin bar copper grids (Agar Scientific, Stansted, UK) and stained in 2% uranyl acetate in 70% ethanol for 20 min, followed by 0.4% lead citrate dissolved at high pH in distilled water for 5 min. They were examined in a JEOL 1230 TEM operated at 100 kV accelerating voltage. Digital images were acquired in using a Megaview III (SIS systems, Olympus Microscopes Ltd) and the height and width of stereocilia were measured as previously described[71]. Semi-thin sections (250–270 nm) of the organ of Corti were obtained in a radial plane and examined unstained in a JEOL 1230 TEM operated at 100 kV. The height and width was measured using imageJ (NIH). The heights of first row IHC stereocilia (s1) were measured only when 90–100% of their length from the distal tip lay within the section (confirmed by examination of the adjacent serial section). The full length of second row (s2) stereocilia was readily captured in these sections and was measured to the maximum distal tip including the bevelled portion where present. The width was measured at half maximum height of each stereocilium. For TEM, $\geq 3$ mice were processed for each set of experiments and for each genotypes. Images were taken from the same region (around 12 kHz) used for the electrophysiological recordings.

**Statistical analysis**. Statistical comparisons of means were made by Student's two-tailed $t$ test or for multiple comparisons, analysis of variance (one-way and two-way ANOVA followed by Bonferroni's test) were applied. $P < 0.05$ was selected as the criterion for statistical significance. Mean values are quoted in text and figures as means ± SEM. apart those referring to ABR and TEM measurements, which are reported as means ± SD. Only mean values with a similar variance between groups were compared. Electrophysiological recordings in which the IHC leak

conductance, normally calculated between −84 and −94 mV, was significantly larger than that previous reported for both immature and adult cells[14] or with a residual series resistance ($R_s$) after compensation > 5 MOhm were excluded from the analysis. Animals of either sex were randomly assigned to the different experimental groups. No statistical methods were used to define sample size, which was defined based on previous published similar work from our laboratory. The majority of the experiments were performed blindly to animal genotyping.

## Data availability

The data supporting the findings of this study are available within the paper and its Supplementary Information files and from the corresponding author upon reasonable request.

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

## Acknowledgements

We thank J. Pass and L. Preite for generating the *Ush1c* and *Myo7a* tm1c allele conditional knockout mice, the Sanger Institute Mouse Genetics Project and EUCOMM for providing the *Myo7a* and *Ush1c* tm1a alleles; Neil Ingham for advice on ABR measurements; Maria Pakendorf for helping with the genotyping and Michelle Bird for

her assistance with the transgenic mouse colonies. This work was supported by the Wellcome Trust to W.M. (102892) and K.P.S. (100669), the MRC (G0100798 and MR/K005561/1) to C.J.K. and a BBSRC Research Studentship to T.R. S.L.J. is a Royal Society University Research Fellow.

## Author contributions

L.F.C., S.L.J., T.R., K.M.R., A.H., F.C., S.S., K.P.S., A.F., C.P., D.N.F., C.J.K., W.M. contributed to the collection and analysis of the data. W.M. conceived and coordinated the study. L.F.C., C.J.K. and W.M. wrote the paper.

## Additional information

**Competing interests:** The authors declare no competing interests.

