## [Peer Review File · Nature Communications]

Corns et al., “The mechanotransducer current is required for...”

In this manuscript the authors describe their studies of mice in which proteins necessary for mechanotransduction are genetically reduced: protocadherin, harmonin and myosin 7A. Each of these has somewhat different effects on the development and maintenance of the mechanotransducer. But in all, their effects are consistent with the central hypothesis that normal levels of excitation due to cationic flux through open transducer channels are required for the acquisition and maintenance of basolateral membrane channels and arrangement of synaptic contacts of the hair cell. This central message, and the data provided to support it, makes this an interesting and important contribution to our knowledge of sensory hair cell maturation. The significance is made still greater by their fascinating observation that the loss of mechanotransducer function causes adult hair cells to revert to an immature physiological state, including the return of efferent inhibitory synapses that exist temporarily during the first two postnatal weeks, echoing a finding for age-related hearing loss in mice. It remains to be determined how these changes affect hearing, but one must assume that these make as-yet undefined, and probably substantial, contributions to hearing loss. The work is thorough and very well done overall, and generally well-written. The basic discoveries are fascinating, and the implications for human health striking. General and specific comments follow.

‘Spontaneous activity’ – throughout the authors make a convincing argument that hair cell depolarization and calcium spiking is necessary for maturation of ion channel expression and synaptic arrangements. While this may be case, all of their results, with one exception (Fig 4D - transient reduction calcium causes depolarization and action potential generation) the sample recordings are from hair cells that are silent until depolarized by current injection. Is it possible to demonstrate recordings from hair cells that show ongoing, unstimulated action potential firing? Alternatively, is it known whether activity in the VIIIth nerve of their mouse models is altered? Presumably it should be absent, according to their hypothesis. This query is motivated in part by work from others (e.g., Tritsch NX, Yi E, Gale JE, Glowatzki E, Bergles DE. The origin of spontaneous activity in the developing auditory system. *Nature*. 2007 Nov 1;450(7166):50-5. PubMed PMID: 17972875.) showing that hair cells and afferent fibers generate action potential bursts in *ex vivo* cochlear tissues where perilymph like saline (low potassium, high sodium, mM calcium) bathes both apical and basolateral surfaces, so presumably corresponding to little or no mechanotransduction due to strong calcium-dependent adaptation). As the authors know, this bursting ‘spontaneous’ activity is driven by ATP-dependent waves among nearby supporting cells. What do the authors predict? Will there still be bursting activity in cochlear afferents of their mutant mice? Or is hair cell depolarization via mechanotransduction absolutely required for afferent drive to brain? The present work shows convincingly that MET-dependent depolarization is necessary for normal maturation of hair cells, as is well-described in the Abstract and Introduction. However, elsewhere in Results and Discussion the authors seem to conflate this phenomenon with the role that spontaneous activity in the periphery plays in driving the refinement of central circuitry. This may not be their intention, but perhaps the distinction between hair cell development, and circuit development could be clarified. Perhaps these have different requirements for degree and/or pattern of ‘spontaneous’ activity.

At several points in the paper the authors state that hair bundles are of normal appearance. It might be best to qualify this statement with “...of normal appearance as observed by DIC microscopy”.

Lines 154-156 confusing language “since the estimated... the application ...increased the resting MET...” clarify somehow. ‘the estimated endolymphatic calcium in vivo is 0.3 mM, at which concentration adaptation of MET channels should be much reduced. Correspondingly, application of 0.3 mM calcium increased the resting MET current...’

Lines 162-164. “IHCs would remain silent”??? unless driven by ATP waves from surrounding supporting cells.

Line 196. But in addition (not and)

Lines 289-293. An example of a complex list of values that are difficult to read through, made more difficult by presenting them in different sequence and pattern from one condition to the next. Try to use one pattern of presentation throughout.

Line 315 – both ACh-evoked current components were present??? Based on the IV only? How can you know this?

Were the synaptic currents characterized any further? Pharmacology would be nice, but at least determine waveform values (tau decay) for comparison to literature.

Lines 412-417. Puzzling that ribbon loss does not occur, and yet this is a common feature of other mechanisms of cochlear loss. Just curious...

Lines 442-443. Noise or aging typically damage outer hair cells. Is there evidence for inner hair cell MET loss in these conditions?

Reviewers' comments:

Reviewer #2 (Remarks to the Author):

This manuscript provides interesting data on the remarkable activity-dependent postnatal plasticity in the auditory periphery. It has been known for a while that mutations in the genes encoding essential proteins for the mechanosensory function of the auditory hair cells (e.g. TMC1 or myosin VI) not only cause deficiencies in the hair cell mechano-electrical transduction (MET) but also disrupt normal maturation of hair cells' basolateral K⁺ conductances. In this study, Corns and co-authors expanded these observations on mice carrying mutations in the tip link component, PCDH15. Most importantly, they demonstrated that genetic knockdown of essential stereocilia proteins harmonin and myosin VII during postnatal development reverts normal acquisition of basolateral K⁺ conductances in the cochlear inner hair cells (IHCs) into immature stage and results in re-appearance of immature cholinergic efferent innervation of these cells. The authors also observed similar reversal to immature IHC basolateral K⁺ conductances in vitro after several days of culturing of gerbil adult organ of Corti explants. Although it is not yet clear whether all these effects are due to the loss of MET current as proposed by the authors (see below), the study does demonstrate a hitherto unknown type of activity-dependent plasticity in the auditory periphery.

Major comments:

The major problem of the current version of the manuscript is its pre-determination to the idea that the observed phenomena are due to the loss of MET current. Personally, I do like this hypothesis very much. However, all reported experiments are only consistent with this hypothesis and none of them tests it directly. The authors implicitly argue that a common abnormality in their experimental systems (PCDH15 mutants, harmonin and myosin VII conditional knockouts, and cultured preparations) is the loss of resting MET current and, therefore, the observed changes in IHC function must be due to the loss of MET current. However, one could argue that the hair cells slowly degenerate in all these experimental systems and, therefore, there are many other abnormalities that may be common to ALL these systems, such as endoplasmic reticulum stress, mitochondrial abnormalities, reactive oxygen species production, apoptotic signaling, etc. The time course from the beginning of cre expression at P4 to the earliest observed changes in K⁺ conductances at P15 (Fig. 3) allows multiple secondary effects unrelated to MET current changes. The authors missed an opportunity to test their hypothesis directly in the cultured preparation (see next paragraph). Without direct experiments, the title should be changed to something like, "Stereocilia proteins are required to maintain the identity of mature mammalian auditory inner hair cells and their efferent innervation" and the data presentation should not be biased by the MET current idea, which could be brought up only in Discussion and only as a likely hypothesis.

Alternatively, the authors may test directly their hypothesis by blocking MET current in the cultured gerbil organ of Corti preparations. If the authors' idea is right, then (based on the data of Fig. 6f,i) relatively non-toxic MET channel blockers such as curare, amiloride, or benzamil should result in accelerated reversal of basolateral K⁺ conductances to immature state within a day or two. The "blocker – no blocker" effect should be especially pronounced in 40 μ M of external Ca²⁺.

The exact mechanism of MET current reduction in myosin VII conditional knockouts is unclear. The myosin VII antibodies are sensitive enough to detect substantial downregulation of the protein at P10 (Fig. 4c). Yet, the changes in MET current are observed only at P16 (Fig. 5b). Perhaps, it's better to admit that we don't understand very well how exactly the loss of myosin VII results in MET disruption rather than hand-waive on the potential presence of residual myosin VII. Also, SEM images on Fig. 5d show only the lack of apparent structural abnormalities. The most interesting analysis of tip link structure and distribution is likely to be very complicated at this age.

Finally, neither this study nor the original report on Myo15-cre mice (Caberlotto et al., 2011) show

the essential controls in Myo15-cre+/- mice. It is not uncommon to see the toxicity of cre expression alone.

Minor comments:

Line 100, reference 19: Did you mean Roux, Human Molecular Genetics, 2009? Reference 19 doesn't show "failure in the acquisition of the mature IHC current".

Lines 409-410: The idea of complete Ca²⁺ clearance from stereocilia by PMCA2 before entering the hair cell body was developed for a frog succulus hair bundle. Experimental data from Fettiplace group show that it might be different in mammalian hair cells – Ca²⁺ may reach at least mitochondrial belt beneath the cuticular plate (Beurg et al., J Neurophysiol, 2010).

Line 443: What are the evidence that ageing abolishes normal MET current? I believe it's at least very controversial. I would remove ageing from this statement.

Legend to Fig.1, line 781: Legend mentions "wild type" while the figure indicates Pcdh15 heterozygous animals.

Figure 1: These data require statistical analysis. I could not find any statistics for these experiments.

Figure 5g and Supplementary table 2: There is a substantial decrease of IHC capacitance in the conditional knockouts. How do the authors explain it? Do the cells shrink? 20-30% changes in cell size should be visible. Could the authors provide at least the light microscopy images of these cells to confirm the lack of a wide-spread degeneration in the cell body?

Fig. 6d: The legend claims that it is a P24+1d culture, while subtitle indicates P24+2d.

Fig. 6e: Likewise, the legend claims that it is a P24+6d culture, while subtitle indicates P24+7d.

Figure 7h-j: Some sort of statistical analysis is also required for these experiments.

Corns et al. '**Mechanotransduction is required for establishing and maintaining mature inner hair cells and regulating efferent innervation**' – submission of the revised version of manuscript NCOMMS-17-19604.

Replies to Reviewers Comments

Reviewer 1

In this manuscript the authors describe their studies of mice in which proteins necessary for mechanotransduction are genetically reduced: protocadherin, harmonin and myosin 7A. Each of these has somewhat different effects on the development and maintenance of the mechanotransducer. But in all, their effects are consistent with the central hypothesis that normal levels of excitation due to cationic flux through open transducer channels are required for the acquisition and maintenance of basolateral membrane channels and arrangement of synaptic contacts of the hair cell. This central message, and the data provided to support it, makes this an interesting and important contribution to our knowledge of sensory hair cell maturation. The significance is made still greater by their fascinating observation that the loss of mechanotransducer function causes adult hair cells to revert to an immature physiological state, including the return of efferent inhibitory synapses that exist temporarily during the first two postnatal weeks, echoing a finding for age-related hearing loss in mice. It remains to be determined how these changes affect hearing, but one must assume that these make as-yet undefined, and probably substantial, contributions to hearing loss. The work is thorough and very well done overall, and generally well-written. The basic discoveries are fascinating, and the implications for human health striking. General and specific comments follow.

‘Spontaneous activity’ – throughout the authors make a convincing argument that hair cell depolarization and calcium spiking is necessary for maturation of ion channel expression and synaptic arrangements. While this may be case, all of their results, with one exception (Fig 4D - transient reduction calcium causes depolarization and action potential generation) the sample recordings are from hair cells that are silent until depolarized by current injection. Is it possible to demonstrate recordings from hair cells that show ongoing, unstimulated action potential firing? Alternatively, is it known whether activity in the VIIIth nerve of their mouse models is altered? Presumably it should be absent, according to their hypothesis.

Pre-hearing IHCs only fire spontaneous AP activity *in vitro* during the first postnatal week. This is because the increased expression of the hyperpolarizing K^+ current I_{K1} , which mainly occurs during the second postnatal week, shifts the IHC membrane potentials negative to the threshold for APs. However, *in vivo* this membrane hyperpolarization is counteracted by the resting depolarizing MET current, which becomes larger with the maturation of the endolymphatic solution and the appearance of the endocochlear potential.

A few years ago we demonstrated that by re-introducing the contribution of the resting MET current *in vitro*, by using the predicted endolymphatic Ca^{2+} concentration in the pre-hearing cochlea (0.3 mM), IHCs were firing APs spontaneously even during the second postnatal week (Johnson et al., 2012, J Neurosci). In the present study we superfused 0.3 mM Ca^{2+} to mimic the *in vivo* situation and the data are shown not only in **Fig. 4d** but also using 2P Ca^{2+} imaging in **Fig. 4e,f**. We mainly used membrane depolarizations when the aim was simply to corroborate the voltage-clamp experiments e.g. if adult IHCs express immature membrane currents then they might also be able to fire APs. Working with an *in vitro* organ imposes experimental limitations. However, when investigating physiological responses we always try to reproduce the *in vivo* conditions as close as possible. In order to make this important point more accessible to the general readership, we have expanded the text in the Results section of the revised MS (**Pg. 7, ln. 11-21; Pg. 8, ln. 5-7; Pg. 10, ln. 4-8**).

Additional considerations: the majority of the work has been performed from IHCs of $Myo7a^{fl/fl}Myo15^{cre^{+/-}}$ mice, in which pre-hearing development is completely normal, including their ability to fire APs under recording conditions mimicking the *in vivo* environment (**Fig. 4**). Normal cochlear development is also demonstrated by the fact that at P15 ABR recordings are normal and IHCs show a normal complement of membrane ion channels. So, there is no reason to assume that the afferent activity is, in any way, altered. In the Ush1C model, the MET current is lost during immature stages and as such IHCs are electrically silent (apart from the possible ATP-dependent contribution). This will most likely affect/abolish the baseline firing activity of the auditory afferent fibres. Although we appreciate the interest of this reviewer in the afferent fibres, the remit of this work is on the role of the MET channel on IHC development and embarking on afferent recordings will not only be extremely time-consuming but will also be a major departure from the scope of the MS.

This query is motivated in part by work from others (e.g., Tritsch NX, Yi E, Gale JE, Glowatzki E, Bergles DE. The origin of spontaneous activity in the developing auditory system. Nature. 2007 Nov 1;450(7166):50-5. PubMed PMID: 17972875.) showing that hair cells and afferent fibers generate action potential bursts in *ex vivo* cochlear tissues where perilymph like saline (low potassium, high sodium, mM calcium) bathes both apical and basolateral surfaces, so presumably corresponding to little or no mechanotransduction due to strong calcium-dependent adaptation). As the authors know, this bursting ‘spontaneous’ activity is driven by ATP-dependent waves among nearby supporting cells. What do the authors predict? Will there still be bursting activity in cochlear afferents of their mutant mice? Or is hair cell depolarization via mechanotransduction absolutely required for afferent drive to brain? The present work shows convincingly that MET-dependent depolarization is necessary for normal maturation of hair cells, as is well-described in the Abstract and Introduction. However, elsewhere in Results and Discussion the authors seem to conflate this

phenomenon with the role that spontaneous activity in the periphery plays in driving the refinement of central circuitry. This may not be their intention, but perhaps the distinction between hair cell development, and circuit development could be clarified. Perhaps these have different requirements for degree and/or pattern of ‘spontaneous’ activity.

This Reviewer has highlighted a very interesting argument. First of all, the solution used by Tritsch et al 2007 is an artificial cerebrospinal fluid (2.5 mM K⁺; 2.5 mM Ca²⁺), which is different from the perilymph-like solution (5.8 mM K⁺; 1.3 mM Ca²⁺) (see Wangemann and Schacht, 1996). Moreover, Tritsch et al (2007) used an intracellular solution containing 10 mM EGTA, which is 10 times larger than that estimated using perforated patch recordings (1 mM EGTA); the problem of using 10 mM EGTA is that it blocks the SK2 current, which is one of the major player in AP activity (Johnson et al., 2011). Finally, we recently showed that in the absence of connexin 26 & 30, which are required for the ATP-induced Ca²⁺ signals in non-sensory cells, spontaneous activity in IHCs is still present and therefore not driven by ATP (Johnson et al., 2017). However, as this Reviewer has also indicated, ATP released by non-sensory cells exerts a modulatory role (bursting-like firing) on the intrinsic IHC firing.

This reviewer is also correct in assuming that this ATP-induced bursting-like firing would (most likely) be present even in the absence of the MET channels (intrinsically silent IHCs). However, as mentioned in our responses above, immature IHCs from *Myo7a^{fl/fl}Myo15cre^{+/-}* mice, which is the main model used in this study, exhibited normal physiological characteristics, so we have no reason to assume impaired activity in the afferent fibres. In the *Ush1C^{fl/fl}Myo15cre^{+/-}* model, the situation is likely to be different but the aim of this work is not to look for changes in the central auditory pathway.

The role of AP activity in the development of the auditory system is quite complex. Current evidence suggests that early APs in IHCs (mainly first postnatal week) are required for the refinement of the auditory circuitry. Additionally, and different from the visual system, the same AP activity is also used to promote the maturation of IHCs during the second postnatal week (this MS). In the Discussion we tried to integrate these two aspects in order to provide a comprehensive overview of the entire developmental process. However, we understand that this might act as a distraction on the main finding from this study. For this reason we have modified the Discussion to emphasise the difference between the two mechanisms in which AP activity seems to be involved (from **Pg. 16, ln. 15** to **Pg. 17, ln. 23**).

At several points in the paper authors state that hair bundles are of normal appearance. It might be best to qualify this statement with “...of normal appearance as observed by DIC microscopy”.

We have performed some additional SEM experiments to provide a better understanding of the hair bundle structure in post-hearing (P15) control (*Ush1C^{fl/fl}*) and knockout (*Ush1C^{fl/fl}Myo15cre^{+/-}*) mice (**Pg. 7, ln. 8-10**). These new SEM images and the quantification of stereocilia length and width are shown in **Fig. 2e,f**). We have also provided some quantification for the hair bundles of *Myo7a^{fl/fl}* and knockout *Myo7a^{fl/fl}Myo15cre^{+/-}* P30 mice (**Fig. 5d**).

Lines 154-156 confusing language “since the estimated... the application ...increased the resting MET...” clarify somehow. ‘the estimated endolymphatic calcium in vivo is 0.3 mM, at which concentration adaptation of MET channels should be much reduced. Correspondingly, application of 0.3 mM calcium increased the resting MET current...’

This section has been re-written (**Pg. 7, ln. 11-21**).

Lines 162-164 “IHCs would remain silent”??? unless driven by ATP waves from surrounding supporting cells.

Changed (**Pg. 8, ln. 15-17**).

Line 196. But in addition (not and)

We are not sure we understand this comment. Hair bundle degeneration occurs in several mutant mice with absent or mutated stereociliary proteins. We have changed the original sentence to make this point clearer (**Pg. 9, ln. 13-16**).

Lines 289-293. An example of a complex list of values that are difficult to read through, made more difficult by presenting them in different sequence and pattern from one condition to the next. Try to use one pattern of presentation throughout.

The text has been changed to make it easier to follow (**Pg. 14, ln. 2-9**).

Lines 315 – both Ach-evoked current components were present??? Based on IV only? How can you know this?

In the presence of ACh, depolarizing and hyperpolarizing voltage steps from the holding potential of -84 mV elicited an instantaneous current in wild-type OHCs, which is carried by SK2 channels and nAChRs (Marcotti et al. 2004). *J. Physiol.* 560, 691-708). In the presence of apamin, which is a specific blocker of the SK2 channels, the outward but not the inward current component was abolished (Fig. 6 of the above paper). On the other hand, strychnine removed both components. This pharmacological approach allows the identification of both the SK2 and the nAChRs components of the ACh-activated current. We have now added a few additional details and referred to the above paper (from **Pg. 14, ln. 24** to **Pg. 15, ln. 3**).

Were the synaptic currents characterized any further? Pharmacology would be nice, but at least determine waveform values (tau decay) for comparison to literature.

We have performed additional pharmacological experiments confirming that the ACh-induced inward currents are mediated by efferent fibres activating $\alpha 9\alpha 10$ nAChRs by using the antagonist strychnine ($1 \mu\text{M}$: **Fig. 7k**). We have also included some analysis regarding the frequency and amplitude of the ACh-induced inward currents in adult IHCs from *Myo7a^{fl/fl}Myo15cre^{+/-}* mice and found that the values were not significantly different to those we have previously reported in pre-hearing IHCs (**Pg. 15, ln. 12-18**).

Lines 412-417. Puzzling that ribbon loss does not occur, and yet this is a common feature of other mechanisms of cochlear loss. Just curious...

This is an interesting point. We decided to include a small paragraph in the Discussion to highlight this “discrepancy” and propose a possible reason (from **Pg. 19, ln. 2-7**).

Lines 442-43. Noise or aging typically damage OHCs. Is there evidence for IHC MET loss in these conditions?

While loss of and damage to basal-coil OHCs is always most prominent in noise and aging studies, a degree of IHC loss and damage is usually reported too. For example, Liberman (1987) describes damage to IHC stereocilia of cats following noise exposure and Bohne et al (1990) report an increase of IHCs with ‘disarrayed, absent or fused stereocilia’ with age in chinchillas. These morphological changes to the stereocilia are likely to lead to MET loss. We have now expanded the text to add some of the above evidence, quoting these papers (from **Pg. 20, ln. 2-5**).

Reviewer 2

This manuscript provides interesting data on the remarkable activity-dependent postnatal plasticity in the auditory periphery. It has been known for a while that mutations in the genes encoding essential proteins for the mechanosensory function of the auditory hair cells (e.g. TMC1 or myosin VI) not only cause deficiencies in the hair cell mechano-electrical transduction (MET) but also disrupt normal maturation of hair cells’ basolateral K^+ conductances. In this study, Corns and co-authors expanded these observations on mice carrying mutations in the tip link component,

PCDH15. Most importantly, they demonstrated that genetic knockdown of essential stereocilia proteins harmonin and myosin VII during postnatal development reverts normal acquisition of basolateral K⁺ conductances in the cochlear inner hair cells (IHCs) into immature stage and results in re-appearance of immature cholinergic efferent innervation of these cells. The authors also observed similar reversal to immature IHC basolateral K⁺ conductances in vitro after several days of culturing of gerbil adult organ of Corti explants. Although it is not yet clear whether all these effects are due to the loss of MET current as proposed by the authors (see below), the study does demonstrate a hitherto unknown type of activity-dependent plasticity in the auditory periphery.

Major comments

The major problem of the current version of the manuscript is its pre-determination to the idea that the observed phenomena are due to the loss of MET current. Personally, I do like this hypothesis very much. However, all reported experiments are only consistent with this hypothesis and none of them tests it directly. The authors implicitly argue that a common abnormality in their experimental systems (PCDH15 mutants, harmonin and myosin VII conditional knockouts, and cultured preparations) is the loss of resting MET current and, therefore, the observed changes in IHC function must be due to the loss of MET current. However, one could argue that the hair cells slowly degenerate in all these experimental systems and, therefore, there are many other abnormalities that may be common to ALL these systems, such as endoplasmic reticulum stress, mitochondrial abnormalities, reactive oxygen species production, apoptotic signaling, etc. The time course from the beginning of cre expression at P4 to the earliest observed changes in K⁺ conductances at P15 (Fig. 3) allows multiple secondary effects unrelated to MET current changes. The authors missed an opportunity to test their hypothesis directly in the cultured preparation (see next paragraph). Without direct experiments, the title should be changed to something like, “Stereocilia proteins are required to maintain the identity of mature mammalian auditory inner hair cells and their efferent innervation” and the data presentation should not be biased by the MET current idea, which could be brought up only in Discussion and only as a likely hypothesis. Alternatively, the authors may test directly their hypothesis by blocking MET current in the cultured gerbil organ of Corti preparations. If the authors’ idea is right, then (based on the data of Fig. 6f,i) relatively non-toxic MET channel blockers such as curare, amiloride, or benzamil should result in accelerated reversal of basolateral K⁺ conductances to immature state within a day or two. The “blocker – no blocker” effect should be especially pronounced in 40 μ M of external Ca²⁺.

As suggested by this Reviewer, we have performed new experiments from P18 and P26 cochlear cultures using 40 μ M Ca²⁺ together with the MET channel blocker d-tubocurarine. The reason for using curare was because it has recently been demonstrated to be non-toxic to IHCs in long-lasting cochlea cultures (Kirkwood et al., 2017) when used at 50 μ M, which should block most of the resting MET current (Farris et al., 2004; Kirkwood et al., 2017). As shown in the revised **Figure 6**, we found that curare caused a rapid reduction of $I_{K,f}$ compared to when IHCs were only bathed with 40 μ M Ca²⁺. Moreover, IHCs also begin to express the inward rectifier K current I_{K1} , which is normally expressed during pre-hearing stages of development. These data are described on **Pg. 13, ln. 6-18**). These results further strengthen our main hypothesis that the MET current is required for the maintenance of the IHC adult identity. For this reason, we decided to keep the original Title of the MS (but shortened to comply with journal style).

The exact mechanism of MET current reduction in myosin VII conditional knockouts is unclear. The myosin VII antibodies are sensitive enough to detect substantial downregulation of the protein at P10 (Fig. 4c). Yet, the changes in MET current are observed only at P16 (Fig. 5b). Perhaps, it’s better to admit that we don’t understand very well how exactly the loss of myosin VII results in MET disruption rather than hand-waive on the potential presence of residual myosin VII. Also,

SEM images on Fig. 5d show only the lack of apparent structural abnormalities. The most interesting analysis of tip link structure and distribution is likely to be very complicated at this age. We have taken on board the Reviewer's comment and expanded the text (from **Pg. 9, ln. 22** to **Pg. 10, ln. 3**). We still believe that a slower turnover for Myo7a could still be a reason for the differences between the two mouse strains since normal Myo7a immunolabelling was seen in at least one mouse out of 4 at P10. However, we have also included the more cautious interpretation of this Reviewer, which is entirely reasonable.

We have also added an SEM image showing the presence of tip links in between the IHC stereocilia of *Myo7a^{fl/fl}Myo15cre^{+/+}* mice (**Fig. 5d**, middle panel). We have also provided some quantification of the length and width of the stereocilia from the SEM images (**Fig. 5d**, bottom panels).

Finally, neither this study nor the original report on Myo15-cre mice (Caberlotto et al., 2011) show the essential controls in Myo15-cre^{+/-} mice. It is not uncommon to see the toxicity of cre expression alone.

This is another valuable point that we have addressed by performing *in vivo* ABRs (P75: **Supplementary Fig. 5**) and patch clamp recordings (P99: **Supplementary Fig. 4**) from *Myo15-cre^{+/-}* mice. These new data are described on **Pg. 11, ln. 15-17; Pg. 12, ln. 3-4**.

Minor comments

Line 100, reference 19: Did you mean Roux, Human Molecular Genetics, 2009? Reference 19 doesn't show "failure in the acquisition of the mature IHC current".

This is not correct. Figure 4 in Roux et al., 2009 (Hum Mol Gen 18, 4615-4628) shows that mature IHCs from *Myo6^{sv/sv}* fail to express the adult-like currents ($I_{K,r}$, $I_{K,n}$) and instead retain an immature phenotype including the ability to fire action potentials, which is in line with the statement provided in the submitted paper. For this reason, we did not modify the original text.

Lines 409-410: The idea of complete Ca²⁺ clearance from stereocilia by PMCA2 before entering the hair cell body was developed for a frog succulus hair bundle. Experimental data from Fettiplace group show that it might be different in mammalian hair cells – Ca²⁺ may reach at least mitochondrial belt beneath the cuticular plate (Beurg et al., J Neurophysiol, 2010).

The Beurg et al (2010) study states that the PMCA2 Ca²⁺ pumps can clear Ca²⁺ loads 'several times larger than either the resting or maximum MT currents, thus providing a reasonable safety margin.' So, under normal physiological conditions, Ca²⁺ would be fully cleared from the stereocilia. They only observed Ca²⁺ signals in the mitochondrial belt in unphysiological high Ca²⁺ solution (1.5 mM). For the above reasons, we did not change our original statement but we have included the Beurg et al. 2010 paper (**Pg. 18, ln. 17-20**).

Line 443: What are the evidence that ageing abolishes normal MET current? I believe it's at least very controversial. I would remove ageing from this statement.

We agree with the reviewer that aging is mainly affecting OHCs. However, Bohne et al (1990) reported an increased number of IHCs with 'disarrayed, absent or fused stereocilia' with age in chinchillas. These morphological changes to the stereocilia are likely to lead to MET loss. We have now included this reference in the revised MS (**Pg. 20, ln. 2-5**).

Legend to Fig.1, line 781: Legend mentions "wild type" while the figure indicates *Pcdh15* heterozygous animals.

The legend has been changed to indicate that the control recordings are from *Pcdh15* heterozygous mice.

Figure 1: These data require statistical analysis. I could not find any statistics for these experiments. We have now included statistical analysis for the experiments shown in Fig.1 (Pg. 5, ln. 10-18).

Figure 5g and Supplementary table 2: there is a substantial decrease of IHC capacitance in the conditional knockouts. How do the authors explain it? Do the cells shrink? 20-30% change in cell size should be visible. Could the authors provide at least the light microscopy images of these cells to confirm the lack of widespread degeneration of the cell body?

In response to Reviewer 1 we have now provided additional recordings from 7-month old IHCs of control (*Myo7a^{fl/fl}*) and *Myo7a^{fl/fl}xMyo15-cre^{+/-}* mice (Fig. 5g,h). This new data show that IHCs are still present at old ages and are healthy (e.g. normal V_m : Pg. 11, ln. 17-24). Interestingly, we found that the C_m of control IHCs also decreases with age, almost reaching the same values as the *Myo7a^{fl/fl}xMyo15-cre^{+/-}* cells, which instead are maintained at comparable values between 2 and 7 months (Pg. 11, ln. 17-24). In order to provide this reviewer with a visual result, we have taken DIC images of the IHCs at P52 (see Figure below), which show the different size between the two genotypes. However, we believe that quoting C_m values provides a better representation of the cell size since it does not depend upon cell orientation. For this reason, the DIC images are not included in the MS.

Fig 6d: The legend claims that it is a P24+1d culture, while the subtitle indicates P24 +2d. Thank you for spotting this contradiction. The value in the Fig. is correct (P24+2d culture). We have corrected the Figure legend.

Fig 6e: The legend claims that it is a P24+6d culture, while the subtitle indicates P24 +7d. As above - we have corrected the Figure legend (P24+7d culture).

Figure 7h-j: Some sort of statistical analysis is also required for these experiments.

We have performed additional experiments confirming that the ACh-induced inward currents are mediated by $\alpha 9\alpha 10$ nAChRs by using 1 μ M of the selective antagonist strychnine (new Fig. 7k). We have also included some analysis of the frequency and amplitude of the ACh-induced inward currents in adult IHCs from *Myo7a^{fl/fl}xMyo15-cre^{+/-}* mice (Pg. 15, ln. 12-18).

Reviewers' comments:

Reviewer #2 (Remarks to the Author):

In this revised version of the manuscript, the authors addressed satisfactory all my concerns. This is now a very convincing and solid story.

I have only two minor comments to the new data and the text:

Figures 2f and 5d: Correct measurements of stereocilia heights on the scanning electron micrographs require images of the same hair cell from at least two "angles of view". Without that, any numbers could be misleading and small variations in the "view angle" (which is different from the tilt of the stage) could hide the potential changes of stereocilia heights that can occur after loss of transduction current (see Velez-Ortega et al., 2017). In fact, it seems to me that almost all stereocilia of the second row in the illustrated bundle of *Ush1c fl/fl Myo15-cre +/-* mouse are shorter than that in the bundle of a control mouse (Fig. 2f). Therefore, the average stereocilia heights shown on the bottom graphs in Fig. 2f and 5d illustrate only that the bundles are apparently normal. I would just make this qualitative statement and remove height measurements, unless they have been properly performed. In the latter case, the exact measurement procedures have to be described and illustrated.

Lines 410-413: I still do not buy the argument that the Ca^{2+} influx through the MET channels is not involved. Even if Ca^{2+} ions are completely extruded from the stereocilia and not entering into the cell, there might be plenty of Ca^{2+} -dependent signaling pathways initiated in the stereocilia that would affect the cell body (e.g. PIP2, calmodulin, etc).

Reviewer #3 (Remarks to the Author):

For the current manuscript, Corns et al. investigate the intriguing idea that hair cell mechano-electrical transduction (MET) contributes to the maturation of cochlear inner hair cells. This is not a new idea, but to date, has not been thoroughly investigated. To examine this hypothesis, the authors take advantage of several cleverly designed mouse models that have disrupted acquisition of MET or accelerated decay of MET. The authors conclude that resting MET current that flows through channels open at rest is required for inner hair cell (IHC) maturation. Unfortunately, the data do not support this conclusion. This is a significant short coming and limits enthusiasm for the manuscript.

The authors seem focused on the resting current as the key factor, however, with the exception of some circumstantial evidence from just one experiment, they have not demonstrated a direct connection between resting MET current and IHC maturation. Nearly all their manipulations inhibit all current through MET channels, not just the resting current. The authors do convincingly demonstrate that the IHCs that lack MET, have immature properties. However, this could be due to multiple primary or secondary consequences. As such, the conclusions of the manuscript are overstated.

To address the hypothesis that the resting MET current is the key aspect that is lost in the MET-null mutants is challenging, if not impossible. If the authors can devise an experiment in which the resting MET current is abolished *in vivo*, but with MET current otherwise intact, they may be able to test their hypothesis in a more rigorous manner.

In addition, the authors state in the abstract, introduction and results and discussion that mice without MET current re-acquire efferent innervation, but nowhere in the manuscript are there

images that substantiate this claim or show efferent neurons. They show a single physiology trace with strychnine that suggests the presence of Ach release onto IHCs, but without further quantification this is hardly convincing. Thus based on the data shown, the re-innervation conclusion is overstated.

Given these two major shortcomings, the manuscript does not support the stated conclusions and without sufficient supporting data, it falls short of the publication criterion for Nature Communications.

I think they do have some interesting observations that are worthy of attention. If they wish to support the stated conclusions, more direct evidence will be required. Alternatively, if they wish to publish the current interesting observations, more conservative conclusions need to be articulated.

Below are a number of points the authors may wish to consider prior to publication:

Line 83: The current amplitudes given here are meaningless without the voltages at which they were recorded. Please include the voltage. Alternatively, and perhaps more informative, provide the conductance. This needs to be fixed for multiple current values throughout the results section.

Line 97-98: Improper citation. Kawashima et al. (2011) did not demonstrate TMC1 was a pore-forming subunit of the MET channel.

Line 127: How was the fraction of channels open at rest determined? I could not find this info in the results, methods or figure legend. Since resting MET current is an important parameter for this manuscript, this needs to be clarified.

Line 133-136: This sentence does not make sense. Please clarify. In addition, the parameters referenced in this sentence have not been recorded in vivo.

Line 154-156: The authors write: "...the application of 0.3 mM Ca²⁺ increased the resting depolarizing MET current in IHCs from control Ush1cfl/fl mice (Fig. 2i,j), which would be sufficient to depolarize them to their AP firing threshold..." Can the authors demonstrate that this is indeed the case?

Line 181-182: The data in this section do not support the conclusion that the resting MET current is essential for the functional maturation of IHCs. The resting MET current is reduced, but so is the MET current overall. It may be that peak current, calcium influx or some other aspect of MET contributes more to IHC maturation than the resting MET current.

Line 201-202: The data do not demonstrate that the resting MET current is required for the generation of spontaneous APs in pre-hearing IHCs.

Line 206-209: The authors write: "Although IHC depolarization was not completely reversed by the MET channel blocker dihydrostreptomycin (DHS), this was due to the low Ca²⁺ solution additionally affecting the SK2 channels in the basolateral membrane of IHCs through the small hole created to gain access to the cell with the patch pipette". How can the authors be sure this was due to the low Ca²⁺ solution additionally affecting the SK2 channels in the basolateral membrane of IHCs? Referencing their previous work is not sufficient.

Line 279: should read "non-toxic".

Line 352: The authors present no images or other data that adult IHCs without functional MET channels became re-innervated by inhibitory cholinergic efferent neurons. Remove or substantiate this claim.

Corns et al. ‘**Mechanotransduction is required for establishing and maintaining mature inner hair cells and regulating efferent innervation**’ – submission of the revised version of manuscript NCOMMS-17-19604A.

Replies to Reviewers Comments

We thank the Reviewers for their additional comments. Line numbers and Figures refer to the revised version of the MS.

Reviewer 2

In this revised version of the manuscript, the authors addressed satisfactory all my concerns. This is now a very convincing and solid story.

Thank you for helping improve the MS

I have only two minor comments to the new data and the text:

Figures 2f and 5d: Correct measurements of stereocilia heights on the scanning electron micrographs require images of the same hair cell from at least two “angles of view”. Without that, any numbers could be misleading and small variations in the “view angle” (which is different from the tilt of the stage) could hide the potential changes of stereocilia heights that can occur after loss of transduction current (see Velez-Ortega et al., 2017). In fact, it seems to me that almost all stereocilia of the second row in the illustrated bundle of Ush1c fl/fl Myo15-cre +/- mouse are shorter than that in the bundle of a control mouse (Fig. 2f). Therefore, the average stereocilia heights shown on the bottom graphs in Fig. 2f and 5d illustrate only that the bundles are apparently normal. I would just make this qualitative statement and remove height measurements, unless they have been properly performed. In the later case, the exact measurement procedures have to be described and illustrated.

We addressed this point by providing a detailed description of how the measurements were done (Pg. 29, ln. 635-643).

Lines 410-413: I still do not buy the argument that the Ca²⁺ influx through the MET channels is not involved. Even if Ca²⁺ ions are completely extruded from the stereocilia and not entering into the cell, there might be plenty of Ca²⁺-dependent signaling pathways initiated in the stereocilia that would affect the cell body (e.g. PIP2, calmodulin, etc).

As this reviewer is implying, there might be a possibility for Ca²⁺ entering the cell body from the stereocilia, although there is not any published evidence for this. However, we have decided to make a more cautious statement (Pg. 19, ln. 419-422).

Reviewer 3

For the current manuscript, Corns et al. investigate the intriguing idea that hair cell mechanoelectrical transduction (MET) contributes to the maturation of cochlear inner hair cells. This is not a new idea, but to date, has not been thoroughly investigated. To examine this hypothesis, the authors take advantage of several cleverly designed mouse models that have disrupted acquisition of MET or accelerated decay of MET. The authors conclude that resting MET current that flows through channels open at rest is required for inner hair cell (IHC) maturation. Unfortunately, the data do not support this conclusion. This is a significant short coming and limits enthusiasm for the manuscript.

The authors seem focused on the resting current as the key factor, however, with the exception of some circumstantial evidence from just one experiment, they have not demonstrated a direct connection between resting MET current and IHC maturation. Nearly all their manipulations inhibit all current through MET channels, not just the resting current. The authors do convincingly demonstrate that the IHCs that lack MET, have immature properties. However, this could be due to multiple primary or secondary consequences. As such, the conclusions of the manuscript are overstated.

This Reviewer is concerned about our statement for the need of the resting MET current for the normal maturation of IHCs. As described in the Result section, this was addressed by performing several experiments (Figs 1-3, Table 1 & Suppl. Fig. 1) described in the Results section (ln. 74-183). However, some aspects of the complex physiology of the mammalian cochlea were not clearly introduced, which could have led to some confusion. Because the concerns raised by this Reviewer could also be common to other readers, we have taken this opportunity to address at least

some of the crucial aspects in the revised MS (Pg. 9, ln. 183-192), which summarize the points listed below.

There are a number of crucial aspects, highlighted in the Results, that support the conclusion that the resting MET current (only) is crucial for the maturation of IHCs:

- 1) When the MET current was affected by mutations in proteins expressed in the stereociliary bundle (myo7a, myo6, Tmc1: published data; harmonin: **Figs. 2&3 & Suppl. Fig. 1**), including those that are only expressed in the tip links (PCDH15: **Figure 1**), IHCs failed to reach their mature biophysical configuration. As suggested by the Reviewer, these experiments cannot fully discriminate whether it is the resting MET current or the maximum size of the MET current that is important for development (pre-hearing stages). However, there is a crucial aspect of cochlear physiology that this Reviewer should consider – although the MET current can be measured *in vitro* by artificially stimulating the hair bundle, *in vivo* it is only modulated AFTER the onset of hearing (see point 2 below).
- 2) Mice start to hear from postnatal day 10-12 onwards. Before that, mice are completely deaf and no electrical or behavioural responses to sound can be recorded, because the ear canal and middle ear are not yet patent (e.g. *Mikaelian & Ruben, 1965, Acta Otolaryngol. 59,451-461*; *Shnerson, & Pujol, 1982, Dev. Brain Res. 2,65-75*; *Romand, 1983, in Development of Auditory and Vestibular Systems (ed. Romand.) 47-88*; *Ehret, 1983, in Development of Auditory and Vestibular Systems (ed. Romand) 211-237*). The consequence is that the stereociliary bundles are not stimulated and there is no modulation of the MET current before the onset of hearing. However, the resting MET current is independent from sound stimulation because it is determined by the composition of the solution surrounding the stereocilia and the gradual building-up during development of the endocochlear potential. Therefore, the only functional input coming from the MET channels *in vivo* and at pre-hearing stages can be due to the resting MET current, which is lost in the mutants, preventing their IHCs from maturing.

To address the hypothesis that the resting MET current is the key aspect that is lost in the MET-null mutants is challenging, if not impossible. If the authors can devise an experiment in which the resting MET current is abolished *in vivo*, but with MET current otherwise intact, they may be able test their hypothesis in a more rigorous manner.

As explained above, in pre-hearing IHCs the MET channel is not stimulated by sound so the only physiological relevant input to IHCs *in vivo* is the resting MET current.

In addition, the authors state in the abstract, introduction and results and discussion that mice without MET current re-acquire efferent innervation, but nowhere in the manuscript are there images that substantiate this claim or show efferent neurons. They show a single physiology trace with strychnine that suggests the presence of Ach release onto IHCs, but without further quantification this is hardly convincing. Thus based on the data shown, the re-innervation conclusion is overstated.

We are a bit puzzled by this statement, since the Reviewer is only referring to one small part of the data that we showed in **Fig. 7**. The long-lasting trace described by the reviewer (**Fig. 7k**) was added during the revision to address one query from a previous reviewer: confirming that the cholinergic receptors were of the alpha9-alpha10, which are those normally expressed in IHCs.

However, the MS also contains a detailed description of efferent-induced receptor currents (**Figure 6h-j**) and quantification (**Pg. 15, ln.341-346**). The method we have used to investigate and quantify the efferent synaptic activity is that used by all leading laboratories in the field since 2000 (e.g. *Glowatzki & Fuchs, 2000, Science*; *Oliver et al., 2000 Neuron* – and a very long list of additional papers published over the last 18 years). The advantage of the method used is that it allows not only to test the presence of efferent fibres, as can also be done with microscopy (as suggested by this Reviewer), but CRUCIALLY it allows to determine whether the cholinergic efferent fibres are FUNCTIONAL, which is the most interesting and crucial aspect of our work (and which cannot be concluded from microscopical observations).

Therefore, our data provide direct evidence for the presence of efferent input to IHCs in mutant mice; as expected, this is not the case in wild-type control IHCs. There are no other additional experiments we can perform that will address the FUNCTIONAL presence of the efferent system and as such strengthen our conclusion.

Given these two major shortcomings, the manuscript does not support the stated conclusions and without sufficient supporting data, it falls short of the publication criterion for Nature Communications.

I think they do have some interesting observations that are worthy of attention. If they wish to support the stated conclusions, more direct evidence will be required. Alternatively, if they wish to publish the current interesting observations, more conservative conclusions need to be articulated.

As discussed above, our data fully support the claims made in the MS. Moreover, the large majority of the MS (**Figs 4-7 & Suppl. Figs. 2-6**) is focused on the role of the MET current in maintaining the identity of IHCs, about which this reviewer had no concerns. Therefore, we believe that the novel findings provided in this MS are well suited for Nature Comms.

Below are a number of points the authors may wish to consider prior to publication:

Line 83: The current amplitudes given here are meaningless without the voltages at which they were recorded. Please include the voltage. Alternatively, and perhaps more informative, provide the conductance. This needs to be fixed for multiple current values throughout the results section.

The values were present in the Methods section (now: **Pg. 24, ln. 524-529**); we moved them in this part of the MS during the revision in order to comply with the word limits. We have now specified the membrane potentials at which the currents were measured the first time these were mentioned and referred back to the Methods (**Pg. 5, ln. 80-88**).

Line 97-98: Improper citation. Kawashima et al. (2011) did not demonstrate TMC1 was a pore-forming subunit of the MET channel.

We have changed the sentence to better reflect the conclusion of the cited paper “*which is necessary for mechanoelectrical transduction*” (**Pg. 6, ln. 98**).

Line 127: How was the fraction of channels open at rest determined? I could not find this info in the results, methods or figure legend. Since resting MET current is an important parameter for this manuscript, this needs to be clarified.

We have now included a statement in the Methods section (from **Pg. 25, ln. 554 to Pg. 26, ln. 556**).

Line 133-136: This sentence does not make sense. Please clarify. In addition, the parameters referenced in this sentence have not been recorded *in vivo*.

The sentence has been re-worded and the reference to “*in vivo*” has been removed (**Pg. 7, ln. 134-137**).

Line 154-156: The authors write: “...the application of 0.3 mM Ca²⁺ increased the resting depolarizing MET current in IHCs from control Ush1cfl/fl mice (Fig. 2i,j), which would be sufficient to depolarize them to their AP firing threshold...” Can the authors demonstrate that this is indeed the case?

The regulation of AP activity in the immature cochlea has been extensively studied for the last 20 or more years with a wealth of publications on the subject. In one of these papers we demonstrated that APs are present throughout pre-hearing stages in IHCs from CONTROL mice when using experimental conditions mimicking as close as possible those *in vivo* (e.g. body temperature, endolymphatic and perilymphatic calcium concentrations and calcium buffering: Johnson et al., 2012, J Neuroscience).

This Reviewer is requesting us to perform the same experiments in CONTROL Ush1cfl/fl mice, which are from the same mouse strain background (C57B). Since in this MS we have used exactly the same experimental conditions used in Johnson et al. (2012), the rationale for re-doing at least a

years' worth of experiments published in a peer-reviewed paper is not clear to us. This also goes against the 3Rs for the use of animals.

Line 181-182: The data in this section do not support the conclusion that the resting MET current is essential for the functional maturation of IHCs. The resting MET current is reduced, but so is the MET current overall. It may be that peak current, calcium influx or some other aspect of MET contributes more to IHC maturation than the resting MET current.

See major comment above.

Line 201-202: The data do not demonstrate that the resting MET current is required for the generation of spontaneous APs in pre-hearing IHCs.

See minor comment above (Reviewer point: Line 154-156): these information in control IHCs is already published (Johnson et al 2012).

Line 206-209: The authors write: "Although IHC depolarization was not completely reversed by the MET channel blocker dihydrostreptomycin (DHS), this was due to the low Ca²⁺ solution additionally affecting the SK2 channels in the basolateral membrane of IHCs through the small hole created to gain access to the cell with the patch pipette". How can the authors be sure this was due to the low Ca²⁺ solution additionally affecting the SK2 channels in the basolateral membrane of IHCs? Referencing their previous work is not sufficient.

As mentioned above, we do not understand why this Reviewer is requiring us to re-do a large amount of experiments that have extensively been investigated in previously peer-reviewed papers (Johnson et al., 2012 JNeurosci & also Johnson et al., 2011 Neuron). This seems an unreasonable request, also considering that this aspect is irrelevant to the main conclusion of the MS. It was just included to highlight a potentially interesting phenomenon we have previously described in IHCs. As mentioned above, this is also against the 3Rs for the use of animals.

Line 279: should read "non-toxic".

Thank you

Line 352: The authors present no images or other data that adult IHCs without functional MET channels became re-innervated by inhibitory cholinergic efferent neurons. Remove or substantiate this claim.

See major comment above.

Reviewers' comments:

Reviewer #2 (Remarks to the Author):

Although I do like this story and I don't share the skepticism of the other reviewer, I was disappointed by the authors' answers to my last two comments.

Figures 2f and 5d: I have pointed out that, contrary to a common belief, it is not easy to measure stereocilia heights on scanning electron micrographs (SEM) with the accuracy that would allow claiming no changes in mutants as the authors do. I've even pointed to the paper that describes the most accurate technique of such measurements as of today. However, the authors ignored this complexity and provided the following generic description of their measurement, "... hair bundles ... were viewed directly or nearly perpendicular to reduce parallax errors. ... about 50% of the bundles from which we took measurements were not fully, but less than 10 degrees away from, perpendicular. The small error viewing from different angles (~8 -10 degrees) was ignored."

There are number of problems with this description:

1) (Minor) If stereocilia heights were measured from the "nearly perpendicular" images like in Supplementary Fig. 7, then why the images in the main figures of the paper (Fig. 2f and 5d) have very different angles of viewing?

2) (Major) Besides determination of the correct viewing angle, one needs to determine the exact location of the base of a stereocilium. How was it done, if the bases of most stereocilia in the first (tallest) and second row cannot be seen behind the second and third row correspondingly (see for example Supplementary Fig. 7)?

3) (Major) I respectfully doubt the authors' claim on the stereocilia viewing from the angle "less than 10 degrees away from perpendicular". How exactly this angle was measured? Assuming stereocilia perpendicular to the cuticular plate, the depth of viewing (along the reticula lamina) must be no less than $\sim 5 \text{ um}$ (height of the panel) / $\text{Sin}(10^\circ) = \sim 30 \text{ um}$ (Suppl. Fig. 7b, bottom)! Obviously, this was not the case, since there are not too many cells visible behind an inner hair cell.

4) (Major) The staircase arrangement of stereocilia rows within a hair bundle depends on the exact location within the cochlea. The authors never specified the exact location of the cells analyzed with SEM.

Overall, the authors' measurements of stereocilia heights seem to be as imprecise as the measurements decades ago. Since the authors apparently "eyeball" the angle of view in an SEM image, the location of the base of a stereocilium, and the location of a cell along the cochlea, it is highly unlikely that they would be able to detect less than 200 nm changes of stereocilia heights, which are expected in the inner hair cells of mutants with disrupted mechanotransduction (Velez-Ortega et al., 2017). As such, the quantifications in Figures 2f and 5d are misleading – they pretend to be scientific and claim no changes, while the imprecise measurement technique cannot detect the expected changes. Therefore, I would again suggest removing quantifications from Fig. 2f and 5d and making only a qualitative statement on the apparently normal hair bundles.

New lines 419-420: I would like to point out again the extreme weakness of the argument that Ca^{2+} influx through the MET channels is not involved. I do understand that the authors don't believe that this influx can reach the cell body, even though experimental data show that it can (see Beurg et al., J. Neurophysiol., 2010) especially in the first postnatal week when the endolymph composition is still developing and the Ca^{2+} concentration is likely to be very high. However, the authors cannot ignore a possibility of Ca^{2+} binding to the Ca^{2+} -binding proteins (such as calmodulin, calbindin, CaBP1, CIB2, etc) WITHIN a stereocilium, which in turn may

initiate long-range downstream signaling anywhere in the cell.

Reviewer #3 (Remarks to the Author):

The author's reply to reviewer comments does not satisfactorily address the stated concerns and the revised manuscript still includes two major concerns remain. I was somewhat surprised by the augmentative and condescending tone of the reply. I hope the authors will take the concerns to heart and in the spirit intended, to help improve their work and the quality of science published in the journal.

The experimental design is clever and the data are lovely. However, over interpretation is problematic.

1) The onset of hearing at ~P12 in the mouse is not in question: I agree it is well established. What is relevant is the status of IHC bundles and MET channels during the first postnatal week. Hair cell MET channels become functional as early as P1 at the base (Waguespack et al., 2007; Lelli et al., 2009; Kim and Fettiplace, 2013). The authors of the present submission argue that, though MET channels are present during the first postnatal week, there is no gating of MET channels and their only influence is MET current on at rest. I agree that there may be no sound-induced pressure waves that traverse the auditory canal. However, there may be sound-induced bundle deflections evoked via bone conduction. There may also be deflections due to non-sound induced shearing motion between the tectorial and basilar membranes. Furthermore, Denk and Webb (1989) showed that hair cells can detect bundle deflections at the limit of thermal noise. Thus, minute bundle deflections may be occurring, gating MET channels all the time. It is pure speculation, and probably wrong, to suggest there are no hair bundle deflections and no modulation of hair cell transduction channels in vivo during the first postnatal week. Even if they are correct and there are no hair bundle deflections during the first postnatal week, their statement "the only functional input coming from the MET channels in vivo at pre-hearing stages is due to the resting MET current" has not been demonstrated by them or anyone. MET channels are non-selective channels that allow, small and large compounds to permeate (Farris et al., 2004). Any of those permeant compounds, including calcium, sodium and potassium, could provide a relevant input. To suggest the resting MET current is the only functionally relevant parameter is overstated and has not been established. In their mutants, not just resting MET current is abolished, but all of transduction is lost, including responses to transient hair bundle deflections and permeation by a range of molecules. There may be a number of other primary or secondary consequences of the mutations. As such, their conclusion that the only important signal lost is resting MET current remains overstated. For these reasons, I stand by my prior review that their conclusion is overstated and not supported by their data. I again recommend the authors either gather the requisite data or tone down their conclusion.

2) The author's also claim that IHCs have been re-innervated by efferent fibers, but is still not fully supported by the data. I agree the functional data demonstrate the presence of Ach receptors on IHCs, but is a long way from convincingly demonstrating reinnervation. Loss of efferent contacts during normal development and then re-establishment of efferent innervation in the mutant animals needs to be shown histologically to validate this claim.

As suggested during the prior round of review, the authors need to either provide additional data supporting their conclusions or tone down their statements. For both issues, the authors can handle these two major weaknesses with textual changes, which will improve the scientific rigor and impact of the work.

Corns et al. 'Mechanotransduction is required for establishing and maintaining mature inner hair cells and regulating efferent innervation' – submission of the revised version of manuscript NCOMMS-17-19604B.

Summary of major changes in response to the Reviewers Comments

The remaining major points from Reviewer 2 and 3 required performing a substantial amount of electron microscopy experiments and analysis. Therefore, we have contacted a world-expert in the use of electron microscopy in the mammalian cochlea (Prof David Furness, Keele University, UK), who has helped us addressing these last concerns; as such he has been included in the author list.

We thank both reviewers for making us do these additional experiments to further strengthen the MS.

Reviewer 2 (hair bundle):

In order to fully address the comments from this reviewer we have fixed additional samples at P25 for Ush1c and P37 for Myo7a mice. We performed TEM serial sectioning which is the gold-standard technique to measure the height and width of stereocilia. This was also dictated by the fact that our initial analysis using the suggested SEM method (Velez-Ortega et al. 2017), which was done in explanted cochleae from pre-hearing IHCs (P4), proved not viable when applied to adult (>P20) cells, at least in our hands. This was mainly due to the fact that in contrast to the P4 *in vitro* explants (Velez-Ortega et al. 2017) which are allowed to flatten and have shorter stereocilia and no tectorial membrane, our adult samples have the tectorial membrane, are on a curved spiral surface and have substantially taller bundles, which are more prone to disturbance. This prevented reliable measurement of the 'rear' and 'front' views in the same stereocilia using the suggested method. As we are sure this reviewer is aware, sectioning in TEM has allowed us to obtain accurate measures of the stereocilia dimensions, having been established by Furness et al. (*J Neurosci* 2008, 28, 6342-6353; *PNAS* 2013, 110:13898-903) as an appropriate method for measuring length and width of adult IHC stereocilia.

These new TEM experiments, which were obtained from ≥ 3 mice for each genotype, are shown in **Fig. 2f** (described in **ln. 134-152**) and **Fig. 5e** (described in **ln. 251-257**). We now only use SEM images to provide evidence that tip links are present in IHCs from *Myo7aMyo15-cre* mice (**Fig. 5d**), and also to show that in these knockout mice hair bundles retain a staircase structure, which we think is useful for a non-specialized readership (**Figs. 2e and 5d; Supplementary Fig. 1**).

Reviewer 3 (efferent fibres/synapses):

We have now included TEM images showing that, as previously demonstrated (e.g. Simmons et al., 1996), the efferent axo-somatic contacts with IHCs are no longer present in young adult (P20) control *Myo7a^{fl/fl}* and littermate *Myo7aMyo15-cre* mice. These new data are shown in **Supplementary Fig. 7c,d** and described in **ln. 338-342**. These data fully support and are consistent with our previous electrophysiological recordings demonstrating that ACh has no effect on P22 IHCs (**Supplementary Fig. 7a,b**).

We also performed TEM experiments in older mice (P37), a time when our extensive physiological data indicate the return of the axo-somatic efferent onto IHCs (**Fig. 7i-k**). The new data show that indeed the efferent fibres re-innervate the IHCs of the conditional knockout (*Myo7a X Myo15-cre*; **Fig. 7m**) but not control (*Myo7a^{fl/fl}*; **Fig. 7l**) mice. These new data are described in **ln. 366-372**. Based on the additional new data, we can confirm that IHCs are re-innervated by the efferent system in the absence of a function MET channel. These new data have considerably strengthened our conclusion.

Reviewer 2

Although I do like this story and I don't share the skepticism of the other reviewer, I was disappointed by the authors' answers to my last two comments.

Figures 2f and 5d: I have pointed out that, contrary to a common belief, it is not easy to measure stereocilia heights on scanning electron micrographs (SEM) with the accuracy that would allow claiming no changes in mutants as the authors do. I've even pointed to the paper that describes the most accurate technique of such measurements as of today. However, the authors ignored this complexity and provided the following generic description of their measurement, "... hair bundles ... were viewed directly or nearly perpendicular to reduce parallax errors. ... about 50% of the bundles from which we took measurements were not fully, but less than 10 degrees away from, perpendicular. The small error viewing from different angles (~8 -10 degrees) was ignored. As mentioned and explained above, we have fixed new samples and re-done all stereocilia measurements using TEM instead of SEM.

There are number of problems with this description:

1) (Minor) If stereocilia heights were measured from the "nearly perpendicular" images like in Supplementary Fig. 7, then why the images in the main figures of the paper (Fig. 2f and 5d) have very different angles of viewing?

We have re-done all the experiments and analysis using TEM (see above).

2) (Major) Besides determination of the correct viewing angle, one needs to determine the exact location of the base of a stereocilium. How was it done, if the bases of most stereocilia in the first (tallest) and second row cannot be seen behind the second and third row correspondingly (see for example Supplementary Fig. 7)?

We have re-done all the experiments and analysis using TEM (see above).

3) (Major) I respectfully doubt the authors' claim on the stereocilia viewing from the angle "less than 10 degrees away from perpendicular". How exactly this angle was measured? Assuming stereocilia perpendicular to the cuticular plate, the depth of viewing (along the reticula lamina) must be no less than $\sim 5 \text{ um}$ (height of the panel) / $\text{Sin}(10^\circ) = \sim 30 \text{ um}$ (Suppl. Fig. 7b, bottom)!

Obviously, this was not the case, since there are not too many cells visible behind an inner hair cell.

We have re-done all the experiments and analysis using TEM (see above). Supplementary Figure 7 has been removed.

4) (Major) The staircase arrangement of stereocilia rows within a hair bundle depends on the exact location within the cochlea. The authors never specified the exact location of the cells analyzed with SEM.

The location is now included in the new Methods section describing the TEM experiments (ln. 675-677).

Overall, the authors' measurements of stereocilia heights seem to be as imprecise as the measurements decades ago. Since the authors apparently "eyeball" the angle of view in an SEM image, the location of the base of a stereocilium, and the location of a cell along the cochlea, it is highly unlikely that they would be able to detect less than 200 nm changes of stereocilia heights, which are expected in the inner hair cells of mutants with disrupted mechanotransduction (Velez-Ortega et al., 2017). As such, the quantifications in Figures 2f and 5d are misleading – they pretend to be scientific and claim no changes, while the imprecise measurement technique cannot detect the expected changes. Therefore, I would again suggest removing quantifications from Fig. 2f and 5d and making only a qualitative statement on the apparently normal hair bundles.

We believe that the new quantification performed using TEM fully addresses the Reviewer's concerns (see above).

New lines 419-420: I would like to point out again the extreme weakness of the argument that Ca²⁺ influx through the MET channels is not involved. I do understand that the authors don't believe that this influx can reach the cell body, even though experimental data show that it can (see Beurg et al., J. Neurophysiol., 2010) especially in the first postnatal week when the endolymph composition is still developing and the Ca²⁺ concentration is likely to be very high. However, the authors cannot ignore a possibility of Ca²⁺ binding to the Ca²⁺-binding proteins (such as calmodulin, calbindin, CaBP1, CIB2, etc) WITHIN a stereocilium, which in turn may initiate long-range downstream signaling anywhere in the cell.

As suggested, we have changed the text to include the possible involvement of Ca²⁺ in the regulation of gene expression (ln. 438-444).

Reviewer 3

The author's reply to reviewer comments does not satisfactorily address the stated concerns and the revised manuscript still includes two major concerns remain. I was somewhat surprised by the augmentative and condescending tone of the reply. I hope the authors will take the concerns to heart and in the spirit intended, to help improve their work and the quality of science published in the journal.

We are very grateful for the feedback provided by this reviewer, and we apologise if our previous reply was perceived as condescending. In the previous round we felt uneasy with some of the requests, but we now fully understand the constructive nature of the comments, which we have tried to address fully in the revised MS. As mentioned in the introductory paragraph (see above), we performed new TEM experiments to demonstrate the temporal re-innervation of the IHCs by the efferent system.

The experimental design is clever and the data are lovely. However, over interpretation is problematic.

1) The onset of hearing at ~P12 in the mouse is not in question: I agree it is well established. What is relevant is the status of IHC bundles and MET channels during the first postnatal week. Hair cell MET channels become functional as early as P1 at the base (Waguespack et al., 2007; Lelli et al., 2009; Kim and Fettiplace, 2013). The authors of the present submission argue that, though MET channels are present during the first postnatal week, there is no gating of MET channels and their only influence is MET current on at rest. I agree that there may be no sound-induced pressure waves that traverse the auditory canal. However, there may be sound-induced bundle deflections evoked via bone conduction. There may also be deflections due to non-sound induced shearing motion between the tectorial and basilar membranes. Furthermore, Denk and Webb (1989) showed that hair cells can detect bundle deflections at the limit of thermal noise. Thus, minute bundle deflections may be occurring, gating MET channels all the time. It is pure speculation, and probably wrong, to suggest there are no hair bundle deflections and no modulation of hair cell transduction channels in vivo during the first postnatal week.

Even if they are correct and there are no hair bundle deflections during the first postnatal week, their statement "the only functional input coming from the MET channels in vivo at pre-hearing stages is due to the resting MET current" has not been demonstrated by them or anyone.

MET channels are non-selective channels that allow, small and large compounds to permeate (Farris et al., 2004). Any of those permeant compounds, including calcium, sodium and potassium, could provide a relevant input. To suggest the resting MET current is the only functionally relevant parameter is overstated and has not been established.

In their mutants, not just resting MET current is abolished, but all of transduction is lost, including responses to transient hair bundle deflections and permeation by a range of molecules. There may be a number of other primary or secondary consequences of the mutations. As such, their conclusion that the only important signal lost is resting MET current remains overstated. For these

reasons, I stand by my prior review that their conclusion is overstated and not supported by their data. I again recommend the authors either gather the requisite data or tone down their conclusion.

We now fully appreciate the point raised by this Reviewer, who is correct in his/her assumptions. Although our previous reply did not fully address the point raised by this Reviewer, the proposed text was only intended to provide a possible interpretation of the results, and as such it would have been better placed in the Discussion rather than in the Results section. Instead of providing a lengthy and speculative Discussion of all possible scenarios leading to hair bundle deflection during pre-hearing stages for which there is no published evidence, we decided to follow the Reviewer's advice and removed all statements and discussion about the resting MET current in pre-hearing stages from the final version of the MS.

2) The author's also claim that IHCs have been re-innervated by efferent fibers, but is still not fully supported by the data. I agree the functional data demonstrate the presence of Ach receptors on IHCs, but is a long way from convincingly demonstrating reinnervation. Loss of efferent contacts during normal development and then re-establishment of efferent innervation in the mutant animals needs to be shown histologically to validate this claim.

As mentioned in the introductory paragraph above, our new TEM data show that:

- 1) Similar to control mice, IHCs from young adult *Myo7a X Myo15-cre* mice (P20) are no longer directly innervated by the efferent fibre, which instead form axon-dendritic contacts with the afferent fibres as previously demonstrated (e.g. morphology: Simmons et al., 1996; physiology: Glowatzki & Fuchs, 2000). This new information is shown in **Supplementary Fig. 7c.d** and described on **ln. 338-342**.
- 2) TEM images also show that at P37 IHCs from *Myo7a X Myo15-cre* mice were already re-innervated by the efferent system (**Fig. 7m**) but not those from littermate control mice (*Myo7a^{fl/fl}*; **Figure 7l**). These new data are described in **ln. 366-372**.

These additional data are entirely consistent with our electrophysiological experiments and the fact that major re-wiring is occurring in these conditional knockout mice.

As suggested during the prior round of review, the authors need to either provide additional data supporting their conclusions or tone down their statements. For both issues, the authors can handle these two major weaknesses with textual changes, which will improve the scientific rigor and impact of the work.

We believe that the revised version of the MS fully addresses the two remaining criticisms from this Reviewer. In summary, we have:

- 1) removed the speculative paragraph about the contribution of the "resting" MET current in regulating the development of IHCs, which has no bearing on the stated main conclusions of the paper. We now only mention resting MET current for the adult configuration, for which we have provided direct evidence using the cochlear cultures; previously requested from another Reviewer.
- 2) performed new TEM experiments to provide morphological evidence for the axo-somatic efferent re-innervation of IHCs from *Myo7a X Myo15-cre* mice.

REVIEWERS' COMMENTS:

Reviewer #2 (Remarks to the Author):

The authors satisfactory addressed all my concerns in this revised version of the manuscript. Congratulations with the great story!

Reviewer #3 (Remarks to the Author):

The authors have done an admirable job addressing reviewer concerns. The manuscript is much improved and now seems suitable for publication. I congratulate the authors on a nice body of work.

Corns et al. '**Mechanotransduction is required for establishing and maintaining mature inner hair cells and regulating efferent innervation**' – submission of the revised version of manuscript NCOMMS-17-19604C.

Reviewer 2:

The authors satisfactory addressed all my concerns in this revised version of the manuscript. Congratulations with the great story!

Thank you

Reviewer 3:

The authors have done an admirable job addressing reviewer concerns. The manuscript is much improved and now seems suitable for publication. I congratulate the authors on a nice body of work.

Thank you